

# Near-surface thermal stratification during summer at Summit, Greenland, and its relation to MODIS-derived surface temperatures

Alden C. Adolph[1,2], Mary R. Albert[1], Dorothy K. Hall[3,4]

[1]Thayer School of Engineering, Dartmouth College, Hanover, NH, USA
5 [2]Department of Physics, St. Olaf College, Northfield, MN, USA
[3]Earth System Science Interdisciplinary Center / University of Maryland, College Park, MD, USA
[4]Cryospheric Sciences Laboratory, NASA Goddard Space Flight Center, Greenbelt, MD, USA

*Correspondence to*: Alden C. Adolph (Adolph1@stolaf.edu)

**Abstract**

As rapid warming of the Arctic occurs, it is imperative that climate indicators such as temperature be monitored over large areas to understand and predict the effects of climate changes. Temperatures are traditionally tracked using in situ 2 m air temperatures, but in remote locations where few ground-based measurements exist, such as on the Greenland Ice Sheet, temperatures over large areas are assessed using remote sensing techniques. Because of the presence of surface-based 15 temperature inversions in ice-covered areas, differences between 2 m air temperature and the temperature of the actual snow surface (referred to as "skin" temperature) can be significant and are particularly relevant when considering validation and application of remote sensing temperature data. We present results from a field campaign extending from 8 June through 18 July 2015, near Summit Station in Greenland to study surface temperature using the following measurements: skin temperature measured by an infrared (IR) sensor, thermochrons, and thermocouples; 2 m air temperature measured by a NOAA 20 meteorological station; and a MODerate-resolution Imaging Spectroradiometer (MODIS) surface temperature product. Our data indicate that 2 m air temperature is often significantly higher than snow skin temperature measured in-situ, and this finding may account for apparent biases in previous surface temperature studies of MODIS products that used 2 m air temperature for validation. This inversion is present during summer months when incoming solar radiation and wind speed are both low. As compared to our in-situ IR skin temperature measurements, after additional cloud masking, the MOD/MYD11 Collection 6 25 surface-temperature standard product has an RMSE of 1.0°C, spanning a range of temperatures from -35°C to -5°C. For our study area and time series, MODIS surface temperature products agree with skin surface temperatures better than previous studies indicated, especially at temperatures below -20°C where other studies found a significant cold bias. The apparent "cold bias" present in others' comparison of 2 m air temperature and MODIS surface temperature is perhaps a result of the near-surface temperature inversion that our data demonstrate. Further investigation of how in-situ IR skin temperatures compare to 30 MODIS surface temperature at lower temperatures (below -35°C) is warranted to determine if this cold bias does indeed exist.



## 1. Introduction

The Arctic is experiencing warming at a more rapid rate than the rest of the world (Stocker, 2014), but the impacts of this increased temperature extend beyond the polar region. Declining sea ice extent and retreat of glaciers contribute to a powerful ice-albedo feedback that leads to further warming on a large scale. This increased warming leads to declining mass balance of the Greenland Ice Sheet, contributing to global sea level rise. Quantifying current and future ice sheet mass balance remains an active area of research (e.g. Rignot et al., 2011; Rae et al., 2012; Vernon et al., 2013) and is critical to improving

projections of sea level rise. Declining Greenland Ice Sheet mass balance is driven in part by changes in surface energy balance, which drives surface temperature and surface melt (Box, 2013; van den Broeke et al., 2016). Furthermore, the energy balance at the snow surface controls the interactions between the snow surface and the atmospheric surface layer. The net surface energy balance is defined by the net shortwave and longwave radiation, as well as sensible and latent heat fluxes, and heat flux from the underlying snow and ice. The net radiation at the surface affects the stability of the near-surface atmosphere and the

extent to which turbulent heat exchange occurs between the snow surface and the lower atmosphere, impacting both local and regional circulation and climate.

        Surface temperature is a critical component of understanding ice sheet mass balance and for tracking changes in surface energy balance, however making accurate measurements of surface temperature across the vast expanse of the Greenland Ice Sheet over a long period of time is challenging (Reeves Eyre and Zeng, 2017). The installation of automatic

weather stations (AWS) across the ice sheet has begun to provide point meteorological data at many locations through programs such as Greenland Climate Network (GC-Net) (e.g. Steffen et al., 1996; Steffen and Box, 2001; Shuman et al., 2001) and the Programme for Monitoring of the Greenland Ice Sheet (PROMICE) which monitors both skin and air temperatures (e.g. Ahlstrøm, et al, 2008; van As et al., 2011; Fausto et al., 2012) ; however, satellite remote sensing provides the opportunity to collect surface temperature with large spatial coverage and sub-daily to weekly temporal resolution.

"Surface" temperatures in climatological studies often refer to 2 m air temperature (Hudson and Brandt, 2005) as it is a standard measurement at meteorological stations around the globe; however, remotely-sensed surface temperatures from satellite-borne sensors in the cryosphere are the actual skin temperature of the surface at the snow/air interface (Warren and Brandt, 2008). In the polar regions, the high albedo of snow in the visible part of the spectrum, and high emissivity of snow at longer wavelengths often leads to the phenomenon of inversions, where temperature increases with altitude. While often

studied on the scale of tens of meters to kilometres above the snow surface (e.g. Philpot and Zillman, 1970; Reeh, 1989; Kahl, 1990), these temperature gradients have been shown to persist within the lowest two meters above the snow surface (Hudson and Brandt, 2005), which may cause a disparity between the "surface" temperature at 2 m and the actual skin temperature of the snow surface. In validation studies or use of remotely sensed temperatures, this distinction is important. Additionally, these temperature gradients resulting from changes in net radiation have important implications for understanding turbulent

exchange between the snow and the atmosphere, which ultimately affects larger scale circulation.



In the summer of 2015, we conducted a field campaign near Summit Station, Greenland to investigate several methods of determining skin and near-surface air temperatures including use of data from the MODerate resolution Imaging Spectroradiometer (MODIS). We use these data to answer the following questions: a) How do summertime meteorological conditions impact near-surface inversions (beneath 2 m height) on the ice sheet at Summit, Greenland? b) How do MODIS surface temperature products compare to in-situ measurements of temperature, and which "surface" temperature measurements are appropriate for direct comparison with MODIS? c) Can the accuracy of MODIS algorithms to calculate surface temperature be improved through better cloud-masking?

## 2. Background

### 2.1. Surface-Based Temperature Inversions

The presence of surface-based inversions in the hundreds of meters of the lower atmosphere in the polar regions has long been established (Sverdrup, 1926) as a feature that results from low absorption of solar radiation by snow and the high emissivity of snow as compared to the atmosphere. Inversions have been characterized in Greenland and the wider Arctic (Reeh, 1989; Kahl, 1990) as well as in Antarctica (Philpot and Zillman, 1970). Conditions to cause inversions are most frequently met in winter when incoming radiation is low. Surface-based inversions have typically been studied with 2m air temperature as the "base" of the inversion and the height of the inversion extending hundreds of meters into the atmosphere or higher. However, work by Hudson and Brandt (2005) demonstrated the presence of a surface-based temperature inversion below 2 m in the winter of 2001 at South Pole in Antarctica, showing that the largest temperature gradient was in the 20 cm nearest to the snow surface. However, Good (2016) presents measurements of skin temperature and 2 m air temperature from sites around the globe, and find that at their polar sites, these two temperatures generally agree well, with the caveats that there is a reduced amplitude of diurnal cycle temperatures at 2 m and that the agreement is worse during summer due to solar insolation. Unlike in Hudson and Brandt (2005) and this study, data presented in Good (2016) are not from continuously snow-covered sites.

Hall et al. (2008) analysed 2 m air temperature data and skin temperature data from across Greenland and discussed conditions that lead to near-surface thermal stratification over snow-covered areas. Incoming solar irradiance and wind speed are two major controls on thermal stratification. Temperature inversions occur when the incoming solar irradiance is small (i.e. during night) and the snow surface emits longwave radiation; the net radiation at the surface is negative, causing heat transport from the air to the snow surface. The opposite phenomenon of temperature lapse can occur when there is significant incoming solar irradiance resulting in net positive radiation at the surface, with higher temperatures closer to the ground surface and upward heat transport from the snow surface to the air. Strong winds can serve to neutralize these temperature gradients by mixing air masses.

In recent years, studies have been conducted on surface energy balance and near-surface processes in Greenland (e.g. Miller et al., 2013; 2015; 2017; Berkelhammer et al., 2016) and Antarctica (e.g. van As et al., 2005; van den Broeke et al., 2006; Kuipers Munneke et al., 2012). At our study site in particular at Summit, Greenland, Miller et al. (2013) studied the





inversions over two years at but consider the 2 m air temperature to be the base of these inversions, and they did not investigate the surface processes beneath 2 m height. They find that inversions are prevalent in winter months and are less intense during summer months and that the presence of clouds results in weaker inversions. In Miller et al. (2015) the impact of clouds on the surface energy budget at Summit is further investigated, and the warming effect of clouds on 2 m air temperatures is shown in all seasons. Details of the Summit, Greenland surface energy balance are extensively documented in Miller et al. (2017). Berkelhammer et al. (2016) discuss the impacts of the surface-based temperature inversions (with 2 m air temperature as the base) on boundary-layer dynamics, showing that the stability of the atmosphere prevents mixing and ultimately limits accumulation at Summit. These recent studies have investigated near-surface processes in the atmosphere above 2 m at Summit because of the importance of surface energy balance and snow/atmosphere exchange in climate monitoring and ultimately prediction of future change in ice mass balance. However, surface temperature gradients in the lowest 2 meters of the atmosphere, which are most relevant for the remote sensing community and also have important implications for changing ice sheet dynamics, have not been definitively studied at Summit, Greenland.

## 2.2. Remote Sensing of Surface Temperature

There are a number of different remote sensing instruments that measure radiance in the thermal infrared part of the electromagnetic spectrum in order to determine surface temperature, including the Advanced Very High Resolution Radiometer (AVHRR), the Advanced Thermal Emission and Reflection Radiometer (ASTER), the Enhanced Thematic Mapper Plus (ETM+), and the MODIS. The theoretical basis for determining temperature of a snow surface based on measured thermal infrared radiance is described by Hook et al. (2007) and Hall et al. (2008) as follows:

$$Ls_\lambda = \left[ \epsilon_\lambda L_{bb,\lambda}(T) + (1 - \epsilon_\lambda) L_{sky,\lambda} \right] \tau_\lambda + L_{atm,\lambda}$$

where $Ls_\lambda$ is the radiance measured by the sensor on a given satellite, $\epsilon_\lambda$ is the surface emissivity at a given wavelength, $L_{bb,\lambda}(T)$ is the spectral radiance from a black body as a function of temperature, $L_{sky,\lambda}$ is the spectral downwelling radiance from the atmosphere on the surface, $\tau_\lambda$ is the spectral transmittance through the atmosphere, and $L_{atm,\lambda}$ is the spectral radiance upwelling from atmospheric emission and scattering. If emissivity, sky radiance, transmittance, and path radiance are known, surface temperature can be determined through measurements of the radiance at the sensor.

The MODIS instrument produces widely-used land surface temperature (LST), and its products are chosen as the remote sensing product for comparison in this work. This instrument, aboard the Terra and Aqua satellites, has been collecting radiance data from 24 February 2000 to present. The surface temperature products of the Greenland Ice Sheet are used as a baseline to investigate future warming trends (e.g. Hall et al. 2012), to monitor melt events on the ice sheet (Hall et al., 2013), and as input for surface mass balance or snowpack modeling (Fréville et al., 2014; Shamir and Georgakakos, 2014; Navari et al., 2016). A number of validation studies present results acquired over various time scales and in different locations to determine the accuracy of the MODIS surface temperature products in the cryosphere (Hall et al., 2004, 2008; Koenig and Hall, 2010; Westermann et al., 2012; Hachem et al., 2012; Shuman et al., 2014; Østby et al., 2014; Shamir and Georgakakos, 2014; Hall et al., 2015; Williamson et al., 2017). Table 1 provides summary statistics related to the results of many of these




validation studies and is discussed in further detail in the discussion section. Overall, a negative bias is present in nearly all
validation studies, where the MODIS surface temperature is less than the measured ground surface temperature, and this bias
is particularly prevalent at temperatures below -20°C.

Some studies (e.g., Hall et al., 2004; Hall et al., 2008; Shuman et al., 2014) use 2 m air temperature to validate the
MODIS surface temperature products, which may be part of the reason for the biases that are consistently present. Other studies
use thermochrons, either shielded (e.g., Hall et al., 2015) or during darkness (Koenig and Hall, 2010). However, Westermann
et al. (2012) and Østby et al. (2014) both use pyrometers to measure thermal longwave radiation and estimate surface (skin)
temperature, and these studies also find a cold bias in the MODIS surface temperatures. Østby et al. (2014) indicate that this
bias is present at lower temperatures during the winter (and that there is a slight warm bias in the MODIS temperatures during
summer), whereas Westermann et al. (2012) show a cold bias at higher temperatures. Identifying if and when this bias is indeed
present is critical to the use of the MODIS surface temperature products over the ice sheet. A bias in the data can obscure or
alter trends within a dataset. Furthermore, it is possible that a cold bias between 2 m air temperature and skin surface
temperature could be indicative of physical processes of temperature inversion and not any issue of MODIS data validity, and
coupled datasets can be used to further develop our understanding of temperature processes in polar regions.

There are two standard MODIS surface temperature products that may be used to study Greenland surface
temperature: the MOD/MYD11 Collection 6 product and the MOD/MYD29 Collection 6 product, where MOD refers to the
Terra MODIS product and MYD refers to the Aqua MODIS product. The MOD/MYD11 product was developed as a land
surface temperature product (Wan and Dozier, 1996; Wan, 2008, 2014). MOD/MYD29 was developed as an ice surface
temperature product (Key and Haefliger, 1992; Key et al., 1997; Hall et al., 2004 and 2012), and while it is typically not
available on land, it will be available as a special product over the Greenland Ice Sheet after further development. Both
MOD/MYD11 and the preliminary version of the MOD29 special product were compared to our in situ data, and
MOD/MYD11 provided a better match to the data, so we use MOD/MYD11 in the analysis.

The MOD/MYD11 method of surface temperature determination uses radiance in MODIS bands 31 and 32, which
correspond to 11μm and 12 μm, respectively. The algorithm used to estimate temperature is referred to as a "split window"
technique because the differences between the 11μm and 12 μm bands are used to account for atmospheric effects on the
measured radiance. MOD/MYD11 estimates an emissivity value based on land cover, presence of water vapor, and estimated
air temperature near the surface using other MODIS bands. This feature exists because MOD/MYD11 is a global product that
estimates land surface temperature on all types of land cover types. Over snow and ice, this presents very little actual
variability; in all of the data we used, the emissivity in band 32 was 0.990, and in band 31, the emissivity fluctuates between
either 0.992 or 0.994. For cloud masking, MOD/MYD11 uses MOD/MYD35, the standard MODIS cloud mask product. This
product gives a probability that a pixel is clear. MOD/MYD11 masks out anything below 95% probability of a clear pixel.

Previous MODIS surface temperature validation studies have used Collection 5 (C5) products; Collection 6 (C6)
products became available in 2014. Improvements were made in the C6 MODIS product, most notably to rectify degradation
in the calibration of the Terra sensors that was apparent in C5; however the sensor degradation was largely in the visible part



of the spectrum and not in the thermal infrared part of the spectrum used to calculate surface temperature (Lyapustin et al.,
2014; Polashenski et al., 2015; Casey et al., 2017). MOD/MYD11 C6 benefits from improved stability of emissivity values

and improved algorithms to account for viewing angle over its C5 counterpart (Wan, 2014). Additionally, in C6, the calibration
of bands 31 and 32 (used in surface temperature calculation) is improved, resulting in a decrease in measured brightness
temperatures. Furthermore, cloud mask algorithms are improved in C6 (Riggs et al., 2017).

## 3. Methods

**3.1. In-situ Measurements**

To characterize temperatures in the lower 2 m of the atmosphere and on the snow surface skin, an autonomous
temperature measurement station was installed approximately 10 km NNW of Summit, Greenland (indicated on a map in
Figure 1) at an undisturbed snow site for 40 days between June 8, 2015 and July 18, 2015. The following measurements were
made at the station with the sensors indicated:

1.  Snow surface skin temperature using Campbell Scientific/Apogee Precision Infrared (IR) Radiometer [Model: SI-
        111]

    2.  Snow surface skin temperature using two iButton thermochron sensors [Model: DS 1922L, used in Koenig and Hall
        (2010)]

    3.  Temperature above the snow surface at 5cm height and within the snow at the following depths: 0cm, 5cm, 10cm,
15cm, 25cm, and 50cm using type T thermocouples

A schematic of the measurement set up is shown in Figure 2. For all measurements, temperatures are measured every 5 minutes,
then averaged and recorded in 30-minute intervals. The thermochron sensors were placed on the snow surface with a string
tied around the circumference of the sensor and attached to a stake in the snow. The thermochron sensors were a silver color,
and they were not shielded. The sensors were occasionally buried by falling or drifting/blowing snow. From June 8 to 25 the

station was visited and maintained every 2-3 days; between June 26 and July 18, the station was unmaintained, and the
thermochron data for that period are not included in the analysis. Thermochron sensors were factory calibrated within a few
months of deployment.

To measure the snow temperature with depth and in the air above the snow surface, type T thermocouple wires were
fed through hollow white delrin rods approximately 0.5 cm in diameter and 30 cm in length, and the delrin rods were mounted

into a central PVC pipe that was then buried in the snow so that the measurements were at the depths as described above. The
ends of the thermocouple wires were stripped approximately 0.5 cm from the end and twisted several times with pliers; they
were not coated with additional weather-proofing. The thermocouple measurements were calibrated against the Campbell
Scientific SI-111 several weeks before deployment. Although measurements at all depths were collected, the focus in this
current investigation uses only the temperatures measured at 5 cm height above the snow surface.

The Campbell Scientific SI-111 Precision Infrared Radiometer covers the wavelength range from 8 to 14 μm. It has
a stated absolute accuracy of ±0.5°C from -40°C to -10°C, and ±0.2°C from -10°C to 65°C. The sensor was factory calibrated



within several months of its deployment. The sensor was mounted on a horizontal rod extending approximately 60 cm out from the supporting tripod, and the sensor was approximately 60 cm from the surface, pointed directly downward. The field of view of the sensor is 22° half angle, so the legs of the tripod did not affect the measurements.


### 3.2. MODIS Products

The high latitude location of Summit, Greenland puts it within the field of view of the MODIS instruments on Terra and Aqua multiple times each day. To compare in-situ measurements to the temporally coincident MODIS collections, we use swath-level products whose file names contain the UTC time of collection within ±5 minutes. Within each swath, we select

the pixel that has the minimum distance from the latitude and longitude coordinates of our in-situ measurement site. Comparisons between temperatures from the MODIS product and the in-situ measurements that are within 30 minutes of one another are used in the analysis. As skin and near-surface air temperatures can fluctuate within a span of 30 minutes, the non-synchronicity may introduce some error to the comparison, but errors should be random and non-systematic as 30-minute windows of both increasing and decreasing temperature are included in the analysis.

In comparisons of MODIS data to in situ measurements, the bias and root mean square error (RMSE) are calculated as follows:

$$Bias = \frac{1}{n} \sum_{i=1}^{n} y_i - x_i$$

$$RMSE = \sqrt{\frac{1}{n} \sum_{i=1}^{n} (y_i - x_i)^2}$$

where n is the number of observations in the dataset, and x and y are the two datasets being compared. Unless otherwise noted,

all errors are reported as a single standard deviation.

### 3.3. Summit Meteorological Monitoring

Summit Station was the location of the Greenland Ice Sheet Program 2 (GISP2) deep core site and has operated continuously as a year-round station for nearly a decade. NOAA has operated a meteorological station at Summit, measuring

the 2 m air temperature using a shielded Logan PT139 sensor. Additionally, wind speed and incoming solar radiation data were also measured as part of the NOAA station data (NOAA ESRL Global Monitoring Division, 2017). The data provided by NOAA and used in this paper have a one minute temporal frequency, and we take a 30 minute average of the data so that the 2 m air temperature is comparable to the IR skin temperature measurements. Further details of the 2 m air measurements are outlined in Shuman et al. (2014). Additionally, through the Integrated Characterization of Energy, Clouds, Atmospheric

state, and Precipitation at Summit (ICECAPS) project, a number of instruments to monitor cloud, atmosphere, and precipitation were installed at Summit in 2010. One of these instruments is the millimetre wavelength cloud radar (MMCR), a Doppler 35

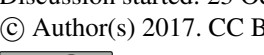



GHz radar that was built in-house and measures reflectivity, mean Doppler velocity, Doppler spectra, and Doppler spectrum width (data available at http://www.archive.arm.gov). More information about the MMCR can be found in Moran et al. (1998). We use MMCR data in this study to detect the presence of clouds and determine the accuracy of the MODIS cloud mask, again
employing the higher temporal frequency measurements and calculating 30-minute averages so that the data are comparable to our in-situ measurements.

## 4. Results and Discussion

The IR measurements and the thermocouple measurements of temperature operated continuously without interruption
during the 40-day campaign. The thermochron dataset ends on June 25, because the thermochrons were not maintained after that date and subsequently became buried in the snow. A time series of the IR skin temperature is presented in Figure 3. The snow skin temperature measured using the IR sensor varied between approximately -34°C and -2°C during the measurement period.

### 4.1. Near-Surface Temperature Measurements

Several different types of sensors were used to measure snow skin and near-surface air temperature during this field campaign in order to compare this study to previous MODIS surface temperature validation studies that used these measurement methods. Figure 4 shows a time series of four different temperature measurements for a subset of the study period. The diurnal cycle of temperature is present in all temperature signals despite continuous solar illumination due to the
changing zenith angle of the sun throughout the day. The difference between the thermochron skin temperature measurement and other near-surface air and skin temperatures is illustrated in Figure 4. Because the thermochron has a mass of several grams and is silver in color, its lower albedo and thermal mass results in heating during peak solar hours. In fact, the thermochron temperature often reports above freezing skin temperatures at times when we are certain that no surface melting was occurring. Because the thermochron was not shielded and has a different albedo than the snow, the thermochron did not
provide an accurate skin temperature record when subject to solar illumination. In Koenig and Hall (2010), temperatures were monitored in the winter during polar night when there was not an issue of solar illumination. Hall et al. (2015) used thermochrons for measurement in March and April in Barrow, Alaska, when there was some sunlight during part of the day. In the Hall et al. (2015) study, both shielded and unshielded thermochrons were used at each site to study and account for issues of solar heating. The shielded thermochrons provided a better match to MODIS surface temperatures than did the non-
shielded thermochrons during sunlight conditions.

Measured IR skin temperature is shown in time series with 5 cm thermocouple temperature and 2 m air temperature for a subset of the study period (approximately 5 days) in Figure 5 to illustrate details of the temperature time series. Temperatures measured 5 cm above the surface produce higher values during the peak sunlight periods of the day than do the 2 m air temperature and the IR skin temperature. Also, their temperatures are not as low as the IR skin temperature at night.
The mid-day difference is likely due to heating of the small amount of exposed thermocouple wire. The wire is silver, and



though it has a very small mass, it has some potential to absorb solar radiation and heat up during peak solar irradiance. The measurements show that thermal stratification causes a difference between snow skin temperature and 5 cm air temperature. When considering only those periods when incoming solar radiation is less than 300 W m$^{-2}$ (to eliminate solar heating effects), near-surface thermal stratification causes temperature differences between the IR skin temperature and the 5 cm thermocouple

air temperature of up to 6.5 °C. These differences are much higher at lower wind speeds; a stronger wind shear allows the system to overcome the stability in temperature and promotes heat flux from the air to the snow surface. Weaker winds cannot overcome the temperature stability so the temperature differences persist. (see supplemental Figure S1).

Thermal stratification in the lowest several meters of the atmosphere is most prominently seen in the difference between 2 m air temperature and IR skin temperature. While 2 m air temperature and IR skin temperature are similar during

peak solar irradiance (Figure 5), there is a larger difference between the two during the night-time, with 2 m air temperature much higher than skin temperature. This is caused by near-surface inversions due to low incoming solar radiation and emission of longwave radiation from the snow surface during the night. This stable condition prevents turbulent heat exchange and allows the inversion to persist. Figure 6a shows a direct comparison between the 2 m air temperature measured at the NOAA weather station at Summit and the in-situ IR skin temperature measured 10km NNW of Summit. The measurements are quite

similar at higher temperatures (above -10°C), but at lower temperatures, there is increased discrepancy between 2 m temperature and snow skin temperature. Figure 6b shows a histogram of the differences between the same 2 m air temperature and IR skin temperature. There is a clear skew in the histogram, indicating that 2 m air temperature is most frequently higher than skin temperature, in both clear and cloudy sky conditions. Figure 7 shows the magnitude of the temperature difference between 2 m and snow skin temperature as a function of concurrent wind speed, with the color of the marker indicating the

concurrent incoming solar radiation. It is clear that increasing wind speed serves to reduce any temperature gradient in the lower meters of the atmosphere, and that at peak solar radiation, there are no inversions present. Specifically, at incoming solar radiation above 600 W m$^{-2}$ or wind speeds greater than approximately 7 m s$^{-1}$, there were not inversions greater than 2°C in the 2 m above the snow surface.

The presence of this near-surface thermal inversion is of particular interest in the context of previous MODIS surface

temperature comparison studies. Several studies have used 2 m air temperature to compare to MODIS surface temperature products (Hall et al., 2004, 2008; Shuman et al., 2014). These studies consistently report a "cold bias" in the MODIS surface temperatures (see Table 1), where MODIS surface temperature is lower than concurrently measured 2 m air temperature. In Shuman et al. (2014), a comparison of MOD29 to 2 m air temperature results in a cold bias of approximately 3°C, and the authors note that the disagreement was larger for lower temperatures. Previous studies acknowledge that near-surface

stratification may be part of the cause of the discrepancy, but also highlight other potential causes such as issues of calibration of the MODIS instruments at very low (<~-20°C) temperatures (Wenny et al., 2012; Xiong et al., 2015), errors in cloud masking, and potential atmospheric interference. The data presented in Figure 6 show that near-surface thermal stratification may play quite a large role in the discrepancies found between MOD29 and 2 m air temperatures (see Figure 1 of Shuman et al. (2014)). Inversions, which are present during periods of lower incoming solar radiation, and thus frequently lower



temperature, result in offsets between skin and 2 m air temperature. Because the MODIS products are a skin temperature (Warren and Brandt, 2008), the difference seen in Shuman et al. (2014) between 2 m air temperature and MODIS temperature at these lower temperatures could in fact be a signature of inversions, which the authors indeed acknowledge but did not have the data to explore. Comparisons of 2 m air temperature to MODIS surface temperature allows us to see how potentially pervasive these inversions could be, though further measurements are needed to determine their presence in non-summer seasons.

Hall et al. (2008) present a figure (their Figure 2) similar to our Figure 6a, in which measured IR skin temperature is plotted vs. 2 m air temperature measured at Summit Station in Greenland from 2000 to 2001. However, they found a consistent offset between 2 m air temperature and skin temperature (of approximately 1°C), a trend that does not vary with temperature. In contrast, our measurements show that the offset is larger at lower temperatures than at higher temperatures and has a much larger magnitude than 1°C; inversions up to 12°C were measured in our data (Figure 6c). In the summer, inversions are present only when solar radiation is low, and therefore temperatures are typically low, so discrepancies between 2 m air temperature and skin temperature only occur during periods of high solar zenith angle. During day time in summer, when there is more incoming radiation and temperatures are typically higher, there is good agreement between measured 2 m air temperature and skin temperature. Because the Hall et al (2008) data span a longer time scale over all seasons, it is possible that the seasonality effects of studying only summer are the root of the differences in our results. However, because inversions are known to be more persistent in the winter than in the summer, we might expect that the trend of larger offsets at lower temperatures would be more pronounced when all seasons are included. Future studies are needed to investigate this discrepancy and determine seasons and conditions under which 2 m air temperature is, or is not, a good proxy for snow skin temperature.

Good (2016) presents results from a study of atmospheric temperatures over many different types of land cover, comparing 2 m air temperature and skin temperature measured from an infrared radiation pyrometer, similar to the instrument used in this study. At polar sites, they find that 2 m air temperature has a reduced diurnal amplitude as compared to skin temperature, but that the two temperatures are generally in good agreement (median differences of ±1.1°C) except in the summer. However, it is unclear if these sites are snow covered in the summer, which may explain why their results differ from ours during this period. Because of the potential issues associated with using 2 m air temperature as a proxy for snow skin temperature, we elect not to compare this to MODIS temperature products. In the following sections, we do compare thermochron data to MOD/MYD11 products, with a distinction between night data and all data because of the issues of heating during peak solar irradiance. We also compare all IR skin temperature to MOD/MYD11-derived surface temperature using swath products when we can match the times of the MODIS and in-situ derived temperatures.

**4.2. In-situ Temperature Comparisons to MODIS Temperature Products**

**4.2.1. Thermochron Temperature Comparison**

It is useful to extract the thermochron data each day from 21:30 UTC - 7:15 UTC (spanning the time around the largest solar zenith angles and therefore lower incoming solar radiation) to compare this "night" data to MOD/MYD11 swath



temperatures. A comparison of all the thermochron data and only "night" data is shown in Figure 8. Using only night data

results in a more favorable comparison to MOD/MYD11 data than using data from the full diurnal cycle. However, even when only the night data are used, the agreement is only fair (RMSE = 4.7 °C). This agreement does appear to be somewhat better during times of higher wind speed (see Supplemental Figure S2). It is likely that during this time of year, there is still too much incoming solar radiation even during high solar zenith angle to use unshielded thermochrons for accurate skin temperature measurement.


### 4.2.2. IR Skin Temperature Comparison

Figure 9 shows a time series of a subset of the measurement period with the 30-minute IR skin temperature measurements overlain with the MOD/MYD11 surface temperature product. MOD/MYD11 does not provide a surface temperature when the cloud mask indicates that there are clouds present, which is why there are some gaps in the data (i.e. at

day 186/187). Most of the time series shown in Figure 9 is during a consistently cloudless period. Terra (MOD) passes over Summit several times in the latter half of the day as temperatures are dropping. Aqua (MYD) passes over Summit as temperatures are typically increasing within the diurnal cycle. The algorithm to calculate temperature from measured radiance is the same in the two different satellites. Figure 9 shows that there is generally good agreement between IR skin temperature and both MOD11 and MYD11 products. This is also evident in Figure 10, where MOD/MYD11 products combine to yield

and RMSE of 1.6°C (n=374) when compared with IR skin temperature, and there is a mean bias of 0.7±1.4°C. In contrast to the results from Shuman et al. (2014), there does not seem to be an increase in the difference between MODIS surface temperature and in-situ temperature as temperatures decrease.

To investigate the root of discrepancies between MODIS surface temperature and IR skin temperature, we consider the sensitivity of the difference between MOD/MYD11 surface temperature and in-situ temperature as a function of the

following parameters: IR skin temperature, solar zenith angle, and sensor viewing angle. These results are presented in Figure 11. The only significant relationship is between temperature difference and MODIS sensor view angle (p = 0.0029). This means that at larger viewing angles, there is a larger difference between the MODIS surface temperature and our measured IR skin temperature, but it does not explain much of the variance, as the $R^2$ value is only 0.02. There is not a significant trend with temperature or with solar zenith angle.


### 4.2.4. Using In-Situ Cloud Data to Improve MODIS Surface Temperature

Using the millimeter cloud radar (MMCR) data from Summit, we identify periods when there were clouds present above Summit Station. While our IR skin temperature measurements were 10km away, we believe that this is still a relatively good proxy for cloudiness, as we resample the data to cover a 30 minute window, so we feel it is more reflective of a larger

area. Figure 12 shows the reduced data, when cloud-affected pixels are removed, for MOD/MYD11. There is an improvement in the RMSE of the data comparison when the cloud-affected data are removed. In determining the strictness of the cloud mask used, there is a trade-off between the number of data points available and the accuracy of the data retrieved. While improving





the cloud mask would improve the data product, it would reduce the amount of measurements available. Østby et al. (2014) also use in-situ cloud data to filter out MODIS surface temperatures that are impacted by the presence of clouds in their study

in Svalbard. Their work shows that the MOD35 cloud mask performs more poorly in the winter than in the summer, so perhaps our results from June and July actually showcase a more favorable measurement period.

## 5. Conclusions

     Data collected during a 40-day field campaign at Summit, Greenland in June and July of 2015 are used to improve

understanding of near-surface temperature on an ice sheet, particularly with respect to MODIS land surface temperature retrieval products. In our comparison of different types of temperature measurement, we find that thermochrons and thermocouple wires, used to measure skin and near-surface air temperature during periods of polar day, can heat up, which may lead to erroneous temperature measurement, and that the thermochrons heated more than did the thermocouples. We also find that at Summit, 2 m air temperature is often significantly higher than skin temperature during the summer months,

particularly at periods of low incoming solar radiation and low wind speed. This near-surface inversion is even present in the 5 cm nearest to the snow surface. This result is important because previous studies that have used 2 m air temperature to validate MODIS surface temperature products have concluded that there was a cold bias in the MODIS data, but our results indicate that the MODIS data may indeed be correct, and the 2 m air temperature is simply not always reflective of skin temperature. Indeed, it is because of the differences between 2 m air temperature and MODIS temperature that we began to

see the pervasiveness of the inversion. We do find that there is a slight cold bias in the MOD/MYD11 surface temperature products as compared to in-situ IR skin temperature, but it is not as large as previous studies have reported, and the RMSE is 1.6°C. The lower RMSE is likely a result of measuring the skin temperature using an IR instrument directly (instead of using 2 m air temperature). During our study period, we measured temperatures down to approximately -30°C. In the future, we plan to extend studies of this type to longer spans of time to determine if these results also are representative of lower temperatures.

Furthermore, the validation presented in this study of the strong correlation between MODIS surface temperature and snow skin temperature in the summer would allow for inversions to be studied more extensively in locations where 2 m air temperature is currently measured. Finally, by using in situ cloud radar data, we confirm, as has been noted in previous studies, that the MODIS cloud mask did not remove all cloud-obscured data from the dataset. When we remove data that were cloud-obscured, the RMSE of MOD/MYD11 improves to 1.0°C. This indicates that improved cloud-masking in the MODIS surface

temperature products could improve the accuracy of the data collected, although it would reduce the total amount of surface temperature measurements available.

## Acknowledgements

Thank you to Vasilii Petrenko for the opportunity to conduct this field work, and to Polar Field Services and staff at Summit

Station for logistic support. Data including 2 m air temperature, wind speed, and irradiance were provided by NOAA's Earth



System Research Laboratory Global Monitoring Division. Cloud radar data were provided by the ICECAPS program. This work was funded by NSF-GRFP 2014186404 and NSF-1506155. Dorothy Hall was funded by NASA-NNX16AP80A.

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

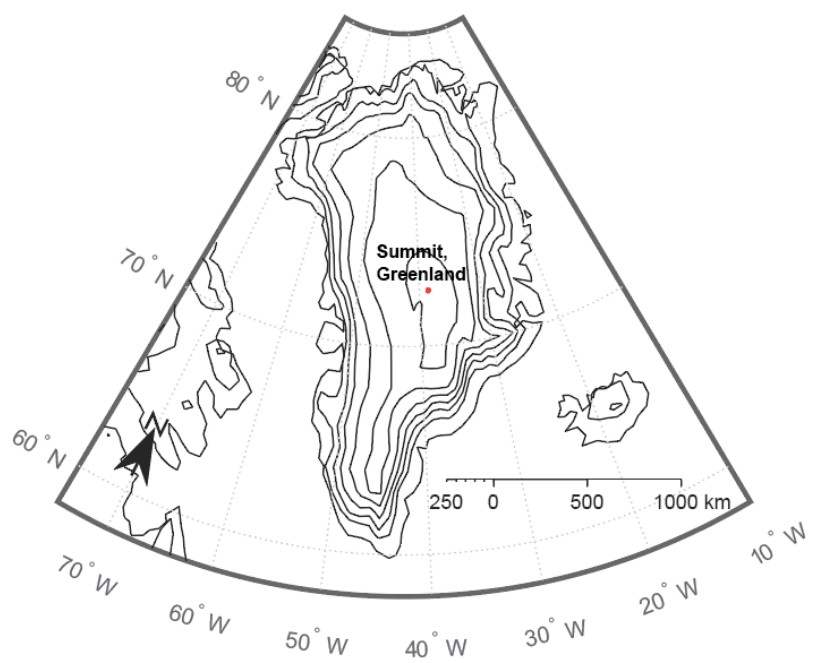

Figure 1: Map indicating the location of Summit, Greenland, the study site for remote sensing and in-situ temperature comparisons. Contour lines represent elevation change of 500m. Latitude and longitude coordinates for the measurement site are 72.65923°N, 38.57067°W.




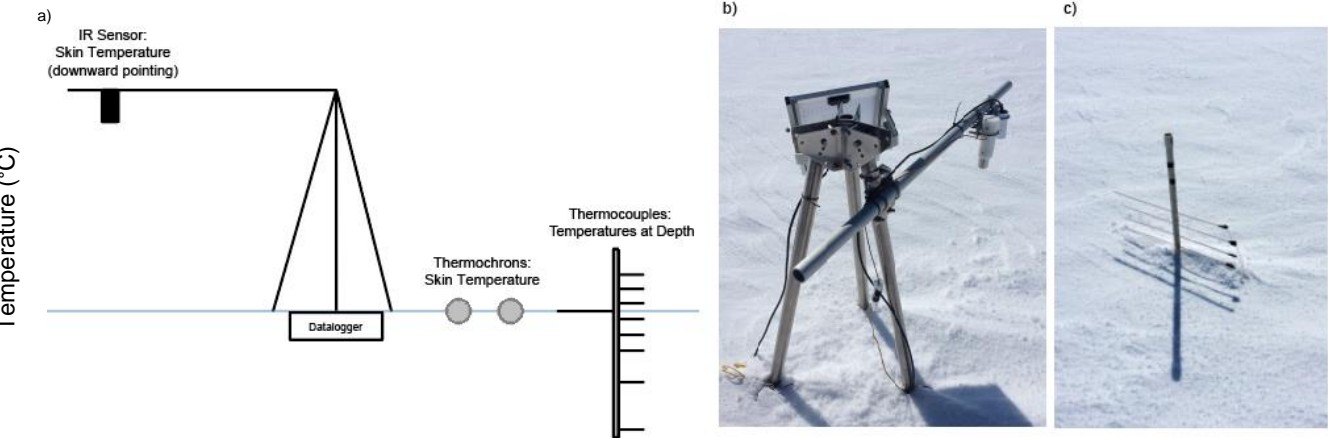

Figure 2: a) Schematic of the types of measurements that were made at the remote station near Summit, Greenland.
Measurements included IR skin temperature, thermochron-measured skin temperature, and thermocouple-measured snow
temperature with depth, b) Image of the IR skin temperature sensor and tripod set up, and c) Image of the thermocouple wire
set up to measure temperature at fixed heights above the snow and depths within the snow.

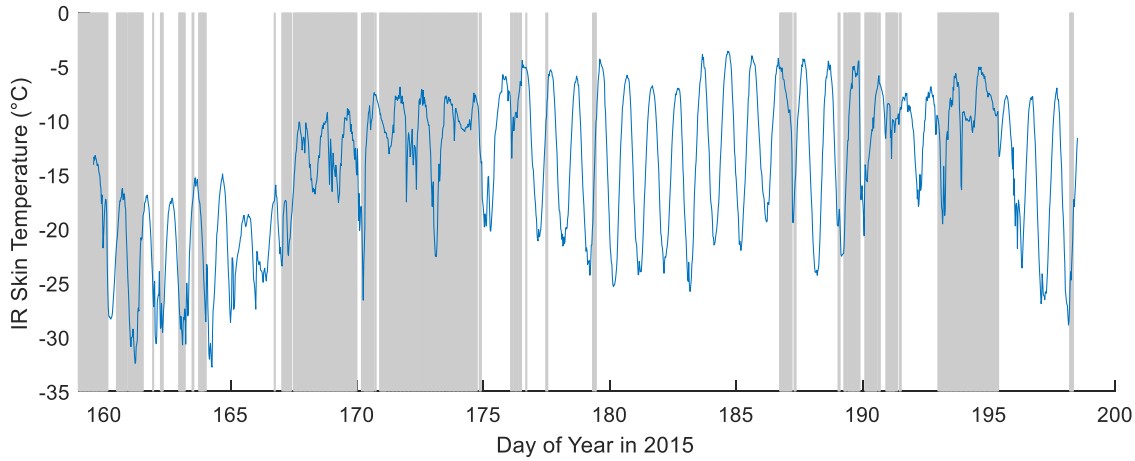


Figure 3: Time series of skin temperature at Summit, Greenland measured with SI-111 IR thermometer (blue). Grey bars
indicate presence of clouds as detected by a millimeter cloud radar at Summit Station.






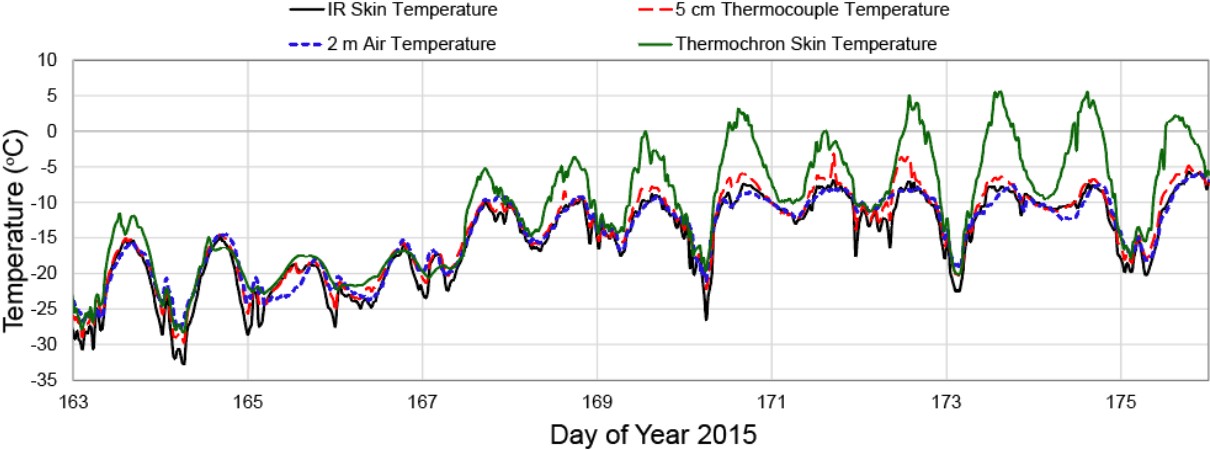

Figure 4: Time series of IR skin temperature, thermochron skin temperature, 2 m air temperature, and 5 cm thermocouple temperature for the duration of the thermochron measurements. In direct sunlight, thermochrons record higher temperatures than IR skin temperatures, 5cm air temperature and 2 m air temperature.


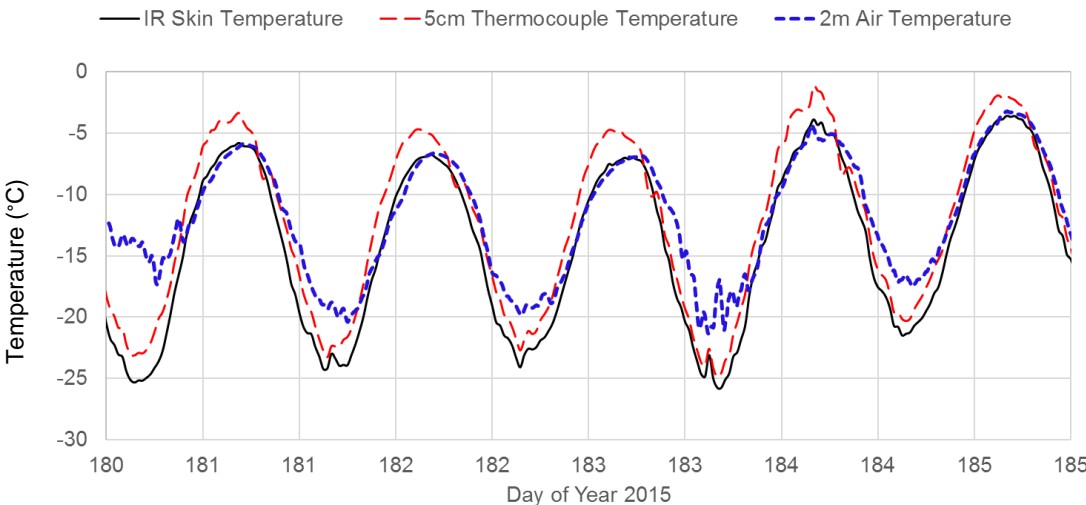

Figure 5: Time series of IR skin temperature, 2 m air temperature, and 5 cm thermocouple temperature during a clear sky period.



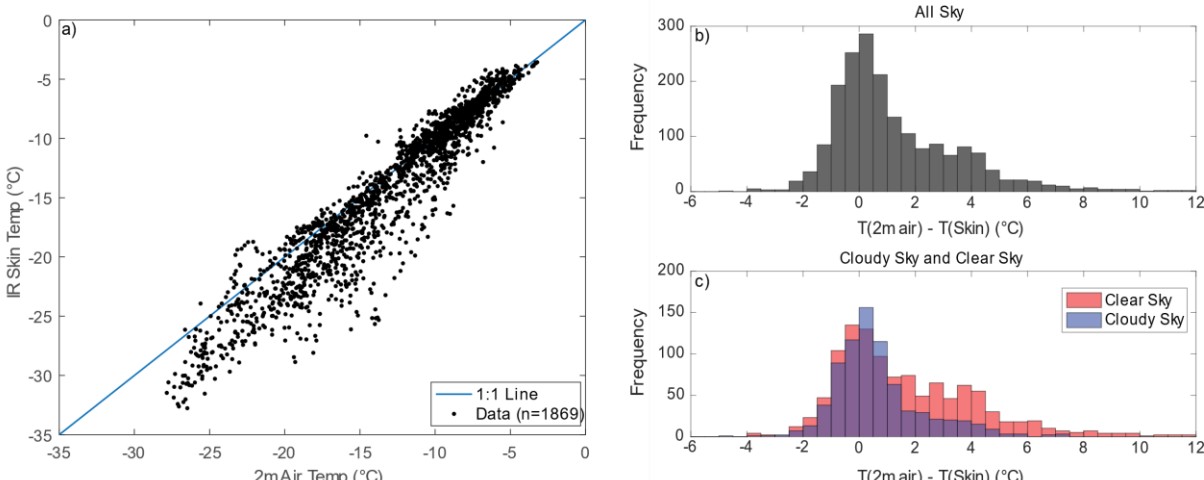

Figure 6: a) Comparison of 2 m air temperature to IR skin temperature at Summit, Greenland during June and July 2015. The
difference between air and skin temperature is largest at lower temperatures. b) Histogram of the difference between 2 m air
temperature and IR skin temperature during the study period in June and July of 2015 at Summit, Greenland during all sky
conditions and c) clear sky and cloudy sky conditions (as detected by MMCR data) separated. The difference is skewed to
positive temperature differences indicating higher air temperatures than skin temperatures.


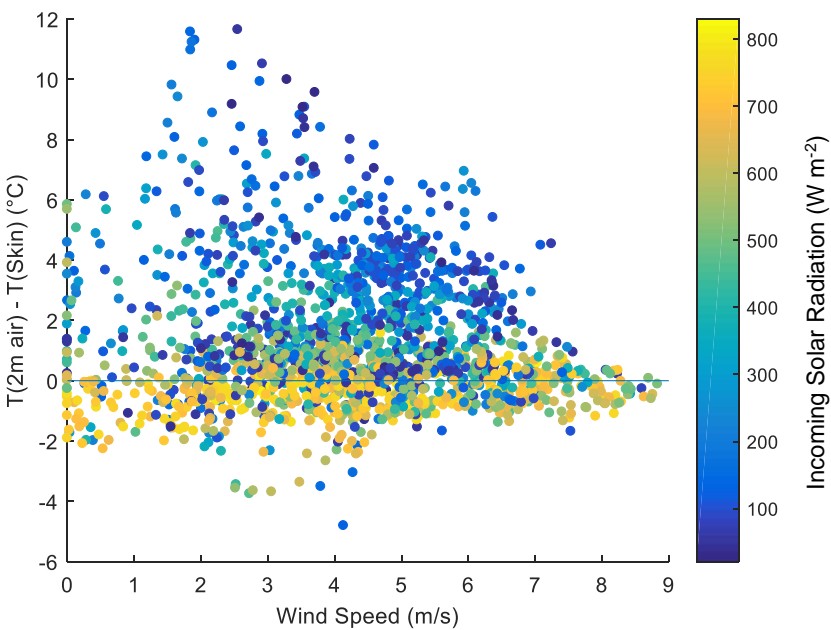

Figure 7: Difference between 2m air temperature and IR skin temperature showing the presence of strong surface-based
inversions at low wind speeds and low values of incoming solar radiation (indicated by the marker colour).






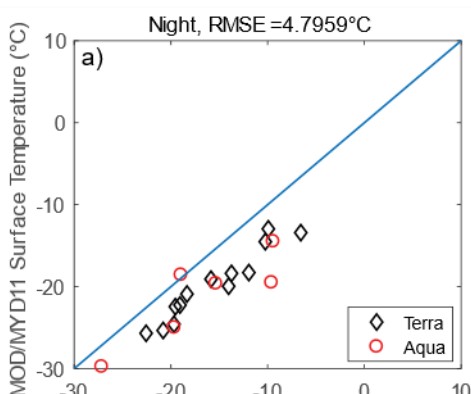 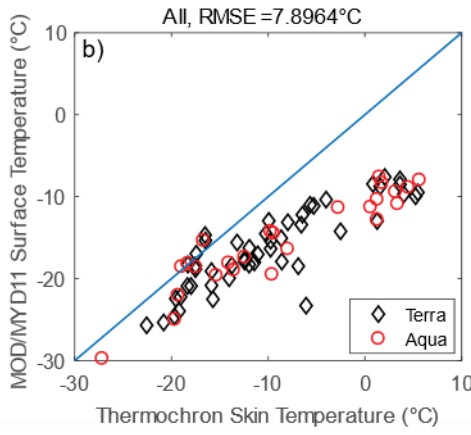

Figure 8: Comparison of thermochron skin temperatures to MOD/MYD11 C6 surface temperature product a) during the night
and b) for all available data. Agreement improves for night-time measurements because thermochrons are not heated by peak
solar radiation, but there is considerable spread in the data.

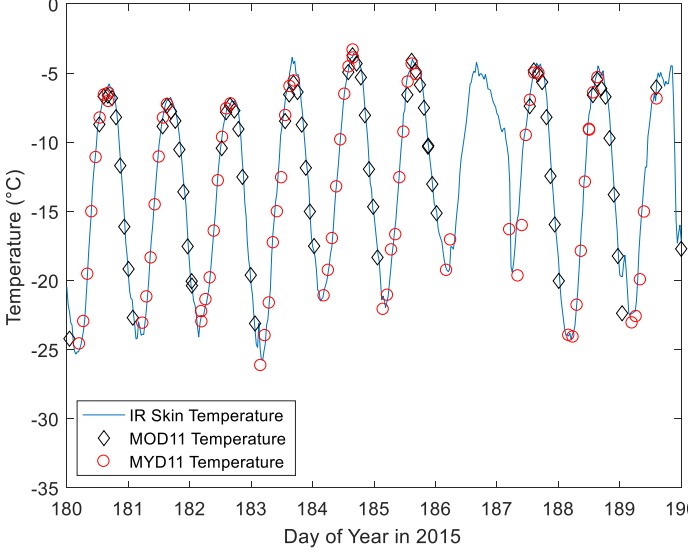

Figure 9: Time series as shown in Figure 3 with only a temporal subset of data presented to clearly show the diurnal cycle of
temperature during fairly clear conditions. Note that the MOD/MYD11 product shows good agreement with IR skin
temperature throughout the diurnal cycle.





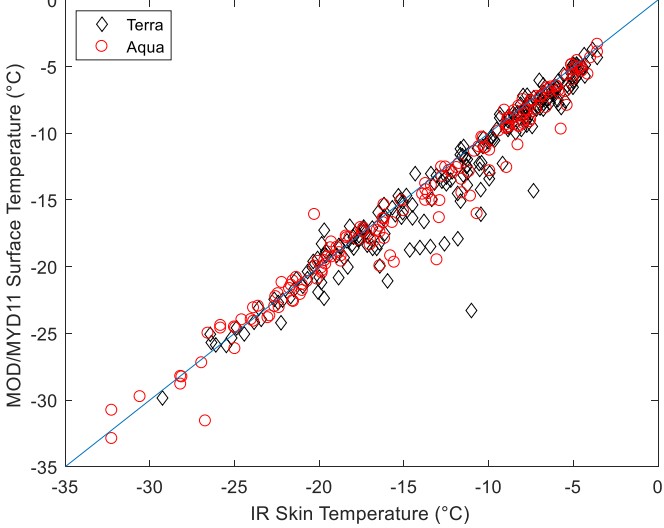

Figure 10: Direct comparison of in-situ IR skin temperature data with MOD/MYD11 C6 surface temperatures. Agreement between satellite and ground-based measurements is quite good (RMSE = 1.6°C, n=374), and there is not a noticeable difference between the performance of the MOD11 and MYD11 temperature products, on the Terra and Aqua satellites, respectively.

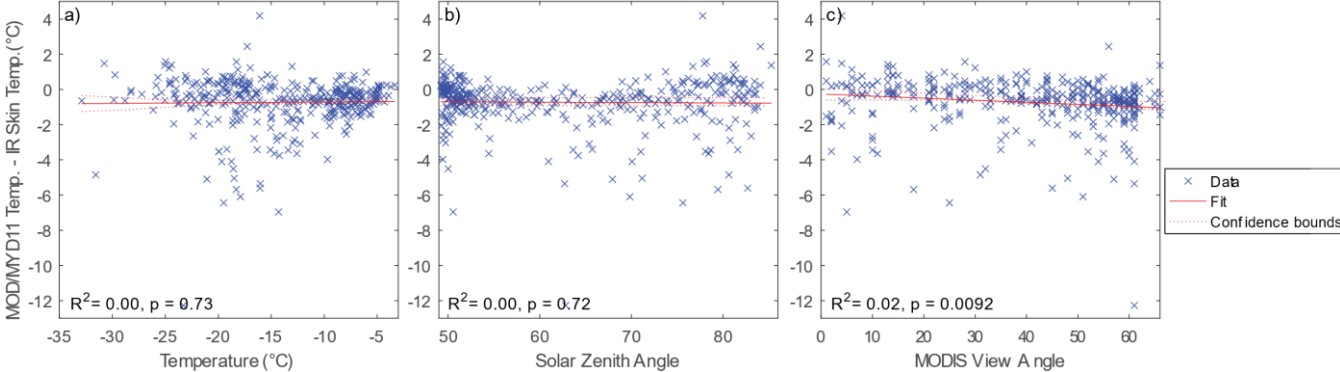

Figure 11: Difference in temperature measured from MOD/MYD11 and in-situ IR skin temperature measurements as a function of a) IR skin temperature, b) solar zenith angle, and c) MODIS viewing angle. The only significant relationship is that the temperature difference is sensitive to the MODIS viewing angle. While the relationship is statistically significant, it is not a strong control on the temperature difference.



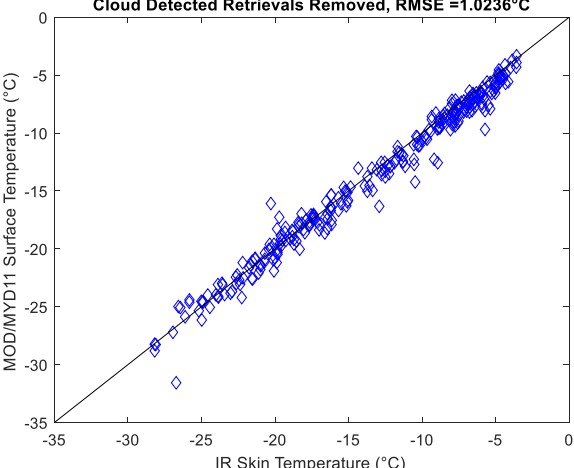

Figure 12: Comparison of MOD/MYD11 to in-situ IR skin temperature after cloud-affected data are removed.

625




Table 1: Summary statistics from recent literature comparing MODIS surface temperature products to in-situ surface temperature measurements

| Study | Location | Temperature Range | Temperature Measurement | MODIS Product | RMSE | Bias |
|---|---|---|---|---|---|---|
| Hall et al. 2004 | South Pole | -70⁰ C to -20⁰ C | 2m Air Temperature | MOD/MYD29 Collection 4 | 1.7⁰ C (n=255) | -1.2⁰ C |
| Hall et al. 2008 | 15 Greenland AWS | -40⁰ C to 0⁰ C | 2m Air Temperature (only during neutral thermal stratification) | MOD/MYD11 Collection 4 | 2.1⁰ C (n=48) | -0.3⁰ C |
| Koenig and Hall 2008 | Summit, Greenland | -41⁰ C to -20⁰ C | Thermochron Skin Temperature | MOD/MYD11 Collection 5 | 3.1⁰ C (n=62) | -3.4⁰ C |
| | | -60⁰ C to -20⁰ C | 2m Air Temperature | MOD/MYD11 Collection 5 | 4.1⁰ C (n=250) | -5.5⁰ C |
| Westermann et al. 2012 | Ny Alesund, Svalbard | -40⁰ C to 0⁰ C | IR Skin Temperature | MOD/MYD11 Collection 5 | | ~-3⁰ C |
| Shuman et al. 2014 | Summit, Greenland | -60⁰ C to 0⁰ C | 2m Air Temperature | MOD29 (Special Greenland Product) Collection 5 | All: 5.3⁰ C (n=2536) Filtered: 3.5⁰ C (n=2270) | ~-3⁰ C |
| Otsby et al. 2014 | Svalbard | -45⁰ C to 0⁰ C | IR Skin Temperature | MOD/MYD11 Collection 5 | All: 5.3⁰ C (n=3941) Filtered: 3.0⁰ C (n=3941) | |
| Hall et al. 2014 | Barrow, Alaska (tundra site) | -42⁰ C to -20⁰ C | Thermochron Skin Temperature | MOD11 Collection 5 | | -2.3±3.9⁰ C (n = 69) |
| | | | | MYD11 Collection 5 | | 0.6±2.0⁰ C (n = 84) |
| This Study | Summit, Greenland | -30⁰ C to 0⁰ C | IR Skin Temperature | MOD/MYD11 Collection 6 (C6) | All: 1.6⁰ C (n=374) Cloud Filter: 1.0⁰ C (n=288) | All: -0.7±1.4⁰ C Cloud Filter: -0.4±0.9⁰ C |
| | | | | MOD11 C6 | 1.8⁰ C (n=207) | -0.8±1.6⁰ C |
| | | | | MYD11 C6 | 1.4⁰ C (167) | -0.6±1.3⁰ C |