# Peer review of "Near-surface temperature inversion during summer at Summit, Greenland, and its relation to MODIS-derived surface temperatures"

_The Cryosphere, 2017_

## Referee Comment (RC1) · Anonymous Referee #1 · 7 Dec 2017

General comments:

The authors present 40 days of near-surface air and surface skin temperature measurements at Summit station Greenland. They analyse the datasets with focus on identifying the timing and magnitude of near-surface thermal inversion. They compare their best measurements of surface skin temperature, from IR upward radiation, to the latest MODIS Land Surface Temperature (LST) and find smaller bias than in other studies validating that product.

The skin temperature is a major variable in the surface energy balance of snow and ice surfaces and has drawn little attention so far. Product such as MODIS LST is a

valuable tool for quantifying that variable but needs ground validation before it is used for the whole Greenland ice sheet. The authors therefor address an important question in a well written study using adequate tools and methods.

However, the study suffers at the moment from the lack of temporal and spatial coverage. Indeed one can ask if these 40 days of observations are representative of the surface conditions on the ice sheet and adequate to validate the MODIS LST as a whole. Especially, it is unfair to compare a 40-days comparison to previous studies that all use multiyear or spatially distributed datasets. As far as I know, NOAA can provide suitable data (IR skin temperature) for a longer period at Summit and PROMICE stations for sites at lower elevation.

Additionally the study could also benefit from more in-depth definition and discussion of all the measurements (f.e. how is calculated MODIS LST, effect of surface emissivity and of the air below the sensor for IR radiation measurements . . .) and concepts (what is skin temperature, how is defined thermal inversion, which indicators and metrics are used to describe it, which are its drivers. . .) that are being used.

Eventually the use of unshielded thermochrons and thermocouples is highly sensitive to radiation (even more during midnight sun period) and should not be used without appropriate segregation of the erroneous periods or correction of radiative heating. At the end of the study, the authors even discard these measurements and only compare MODIS to IR surface radiation. I would recommend greatly minimizing the description of that data and any conclusion that derives from it (f.e. statement regarding inversion in the 5 cm above the surface).

Technical corrections:

l. 19: Define NOAA. Define again in main text.

l.84: Please only mention the conclusions from Good (2016) relative to snowy surfaces.

l. 100: "at but" word missing?

l. 105: It is said" Berkelhammer et al. (2016) discuss the impacts of the surface-based temperature inversions (with 2 m air temperature as the base)". I can see in Berkelhammer et al. (2016) that they have measurements down to 20 cm. Please adjust or justify.

l. 114: "There are", there is

l. 119: The equation points at the various variables that need to be calculated before the surface temperature can be determined. Please give a brief description of how it is being done for MODIS LST and how accurate is that calculation.

l. 126: "its products are chosen as the remote sensing product" rephrase

l. 130: "A number . . ." here validation studies are described. It seems to be the same topic as the next paragraph so consider merging them. Also you discuss validation of MODIS products before you actually describe what these products are and how LST is being calculated. Consider rearranging paragraphs in this section.

l. 134: You mention the recurrent confusion between near-surface, 2 m air temperature and surface skin temperature. Please define your vocabulary at the beginning of the manuscript and then be specific every time temperature is mentioned. Here what does "ground surface temperature" refer to? Check other cases throughout the manuscript.

l. 140: You mention a cold bias in Østby et al. (2014) but it is not reported in Table 1.

l. 144: "A bias in the data can obscure or alter trends within a dataset." Very general statement, consider removing.

l. 145: "Furthermore, it is possible that. . ." at this point, you have not exposed any theoretical (potentially from how MODIS LST is being calculated) or empirical reasons to think that one of the 2 m or skin temperature would match better than the other to the MODIS product. Either add a justification or reference for this hypothesis or move it to what is being interpreted from your validation.

[Figure]

l. 148: What do you mean by "standard"?

l.154: "Both MOD/MYD11 and the preliminary version of the MOD29 special product were compared to our in situ data". If both have been compared and validated then both should be presented. The conclusion that a LST product developed for land performs better than a LST product developed for ice on the Greenland ice sheet is a very important conclusion. Quantifying and localizing the errors in MOD29 could be important for future studies. You can also validate only MOD/MYD11 but then don't mention that you validated MOD29 but not show the result.

l.156: In this paragraph, the authors should provide a clear and concise description on how LST (and all necessary variables such as water vapour, air temperature, emissivity, and cloud cover) are being calculated. It is important so that the reader can be reminded of the assumptions and uncertainties linked to the calculation and to which level of accuracy can be expected from the product.

l. 160: "feature" not clear what it refers to.

l. 161: "Over snow and ice, this..." Not very clear, please rephrase.

l. 165: In this paragraph, please quantify the difference between C5 and C6. More especially, how much of the cold bias seen in validation studies can be explained by the defects of C5 products?

l. 171: "decrease in measured brightness temperatures" define brightness temperature or use vocabulary previously defined. Does this sentence imply that the corrections in C6 would lead to even colder bias if compared to the previously mentioned validation studies?

l. 210: "the pixel that has the minimum distance" a pixel is an area so the station should be located within one at all time. Provide pixel size at some point in the manuscript.

l. 213: "the nonsynchronicity may introduce some error to the comparison" It seems that the IR radiation comes is recorded at 5 min interval and 2 m air temperature at 1

min interval so this nonsynchronicity error could be removed by taking the measurements that are within few minutes of the MODIS acquisition time. Even if it is random noise, removing that error could potentially show better match of MODIS LST with ground measurements.

l. 246 "Several different" redundant

l. 247 "in order to compare this study to previous [. . .] studies" I assume the main goal was to validate MODIS, comparing to other validation studies comes after. Maybe rephrase.

l. 255 "In Koenig and Hall (2010) . . ." the following sentences should be in the methodology where the measurements and their expected limitations are being presented. Additionally Hall et al. (2015) seem to conclude that unshielded thermochrons are subject to measurement error. So why using/presenting that data at all? The use of unshielded thermocouple is subject to the same issue.

l. 270: "differences are much higher at lower wind speeds" Unfortunately, conditions favourable to inversions are also the one enhancing radiative heating of both thermochrons and thermocouple: low wind speeds at 2m imply even lower wind speeds at the surface and will hamper ventilation of the sensors leading to sustained radiation absorption even with low sunlight. Better information should be given to ascertain that this difference is due to inversion.

l. 274 "similar" quantify

l.275 "larger" quantify

l. 280 "increased discrepancy" seems redundant to what is said in the previous sentences. Also quantify here.

l. 281 "most frequently" quantify.

l.289 The following paragraph brings in discussion about MODIS surface temperature

when we are still in the ground measurement section. Consider moving to the next section.

l. 294 "Previous studies acknowledge that near-surface stratification may be part of the cause of the discrepancy" this is very important and has not been mentioned clearly in the introduction. How did these studies arrive to these conclusions?

l. 311 "In the summer..." and following. Can you be more quantitative, f.e. binning observation into night and day time and showing that the mean differences in each group are statistically different. Maybe also find a cut off value in radiation below which you have significantly stronger inversion.

l. 319 You are just repeating what is said about Good (2016) in the introduction.

l. 322 not clear what "median difference" with a +/- sign refers to.

l. 332 Can you specify what is the average solar radiation and zenith angle during that time.

l. 335 Since you aim at validating MODIS LST, better or worse match with it is not a good reason for discarding ground measurements. Consider quantifying the error of the thermochrons using IR temperature instead. If these measurements were not considered reliable enough to validate MODIS LST, why are they being used to quantify the thermal inversion earlier in the study?

l. 335 please give mean bias along with RMSE

l. 350 please give mean bias along with RMSE

l. 356 please state at which viewing angle the error is maximal and quantify that maximum error

l. 360 Why not comparing MODIS to the thermocouple data that you also presented earlier? If, as for the thermochrons it is considered as erroneous measurements it should also be stated and in that case it shouldn't be used for quantifying near surface

inversion earlier in the study. Also consider comparing MODIS with the 2 m temperature over the study period and see if you find same bias as previous validation studies using 2 m temperature.

l.362 Along with the present discussion, it would be important to analyse the performance of MOD35 using the MMCR (false positive, false negative) to draw more broad conclusions (also using findings from Østby et al. (2014)) about when and where MODIS LST might be wrong because of cloud cover.

l. 381 "5 cm nearest to the..." this conclusion comes from potentially heated thermocouple. Needs to be mentioned.

l. 387 "the lower RMSE is likely ..." here you make a hypothesis when you could actually show it. Is your RMSE greater when comparing MODIS to 2 m temperature? Please discuss mean bias along with RMSE.

---

## Referee Comment (RC2) · Anonymous Referee #2 · 12 Dec 2017

General Comments:

The authors present results from a new field campaign near Summit Station, Greenland conducted between 8th June and 18th July 2015. This campaign data is used to investigate near-surface temperature inversions at the site and validate MODIS MOD/MYD11 Collection 6 products. Overall this is an interesting paper which presents original results from a short field campaign as well as from validation of MODIS collection 6 products over Greenland. The manuscript is generally well-written and the methods used are appropriate.

However, I feel that there are issues which need to be addressed by the authors in

redrafting this manuscript. There are also sections of this manuscript which would benefit from a tightening of prose and improvement of structure. Both are elaborated on in the specific comments.

Specific Comments:

It was hard to ascertain the new and original contributions that this manuscript provides to current scientific understanding at first. Outlining these in the introduction or similar would aid reader understanding.

Line 14: suggest "can be assessed" or similar rather than "are assessed" as satellite derived temperatures over land are not in common use for this yet.

Line 53-54: The section starting "however, satellite remote sensing" should mention that the focus of this manuscript is on thermal infrared remote sensing, rather than microwave remote sensing, of surface temperatures. It should also be mentioned that thermal infrared remote sensing observations are affected by cloud cover. The sentence currently gives the impression that the satellite derived surface temperatures are spatially and temporally complete.

Line 60-62: "inversions... which may cause a disparity between the 'surface' temperature at 2m and the actual skin temperature of the snow surface". This sentence currently gives the impression that there is uncertainty about whether inversions cause a disparity between skin and 2 m air temperatures over snow and ice. Yet, the references cited in Section 2.1 note both the presence of inversions in these areas and their effect on temperature stratification. In a related issue, the manuscript could be read as suggesting that the issue of inversions (or other causes of disparity between 2m and skin temperatures) in validation of surface temperatures from thermal infrared remote sensing has not previously been considered. However, various previous studies have noted the issue of the difference between skin and 2 m air temperature over snow and ice surfaces in relation to remote sensing of surface temperatures and their validation. The effect of inversions on the difference between skin and 2 m air temper-

ature over snow and ice surfaces has been noted by e.g. Comiso et al, 2003; Yu et al, 1995. There are also other reasons why differences may be seen between these two temperatures over cryospheric surfaces under clear sky conditions e.g. ice crystal precipitation (Yu et al, 1995) and latent heat effects (Wiese et al, 2015). As a result it is recommended that validation of satellite surface temperatures over land areas is done with situ surface temperatures (ideally from ground-based radiance measurements) if possible (Guillevic et al, 2017). In light of these previous studies and recommendations, this sentence at least should be rewritten and references included in the manuscript to give credit to previous work looking at the issue of the difference between skin and 2 m air temperature over snow and ice surfaces in relation to validation of remotely sensed surface temperatures.

Lines 66-73: Does question b) refer to the specific in situ sensors you use, or to the use of in situ skin temperature data for validation in preference to air temperatures? I could not find these questions referred back to in the results or conclusions. Would suggest removing these or referring back to them later in the manuscript.

Section 2: The content of the background section was informative and interesting. However, in combination with the introduction and methods I found the structure of the manuscript a little hard to follow here. This section would benefit from a rewrite, or a restructuring. Some suggestions follow. The content of the surface temperature inversion section would provide a nice introduction to the issues raised in lines 55-65. The Remote Sensing of Surface Temperatures section mostly focuses on MODIS rather than remote sensing of surface temperatures more generally. The content relating to MODIS products could be moved to Section 3.2 or the section could be renamed.

Line 215 to 220: these metrics are in very common use so the equations do not need stating unless there are notable differences from how they are commonly applied.

Line 241: There is very little discussion relating to figure 3. Reconsider including this figure or provide more discussion of the results shown.

Section 4.1: I found this section quite hard to read and understand. Please rewrite or restructure. Also, please include some values for the differences (bias, RMSE, etc.) noted between the in situ sensor surface temperatures to provide context to previous study comparisons (lines 304-329).

Lines 246-260: Given the issues noted with the thermochrons, please reconsider including these or provide some comment on whether these sensors are suitable for measuring surface temperature. If the authors decide to retain the thermochron analysis, in Figure 4 the difference between the thermochrons and the other sensors increases noticeably after day of year 167 but before they are buried by snow. Please provide some comment on this.

Lines 266-267: "The measurements show". Ambiguous as to whether this refers to the measurements affected most by solar heating or other observations.

Lines 304-318: Did Hall et al. 2008 use the same infrared radiometer as in the study detailed in the manuscript? Table 1 suggests this study only looked at 2 m air temperature.

Line 317-318: Sentence is ambiguous as to whether the need for future studies refers to the results of your study specifically, or in general.

Lines 323-324: Is there publicly available snow depth information at these sites to address this question if it is not included in the paper?

Lines 324-329: Why was the thermocouple data not compared to MODIS? If there is a reason this should be stated.

Lines 353-359: Do the authors have any thoughts on what else could be causing the remaining differences? If so please include this.

Lines 366-368: I think the authors say "improving the cloud mask" when they mean increasing cloud masking strictness which may improve the product but also overflag cloud so that there is loss of un-cloud contaminated data ("reduce the amount of measurements available"). Data loss due to cloud masking, assuming that the pixels removed are genuinely cloud contaminated and the cloud masking is therefore accurate, is not a problem as these pixels will not contain sensible infrared surface temperature estimates. The issue is when there is significant over-flagging of cloudy pixels, leading to loss of non-cloud contaminated data, due to an increase in cloud masking strictness. If so these sentences (and lines 395-396) should be re-written.

Line 384: Do not use the word "correct" here as this suggests that MODIS is perfectly accurate, when actually MODIS data (and indeed any observation) will not measure the true (generally unknown) value of the surface temperature. There are always biases and uncertainties when measuring.

Lines 390-391: Sentence unclear in meaning. Are the authors suggesting using MODIS data and in situ 2 m air temperature to study inversions? If so, are the biases between the MODIS and in situ skin temperatures understood adequately to allow such a study? Lines 353-359 suggest not.

Table 1: This table could do with a little restructuring and/or reduction of text as it is currently a little difficult to read and understand. Also, if this is for studies over land ice only please include this information in the caption.

Technical Corrections:

Line 47: "for understanding ice sheet…" rather than "of understanding ice sheet…"?

Line 53: remove extra space in "Fausto et al., 2012) ; however"

Line 534: remove curly brackets

Figure 1: The location dot is a little small. The north arrow is also a little difficult to see.

Table 1: missing "n=" on last row.

References:

[Figure]

Comiso, J. C. (2003). Warming trends in the Arctic from clear sky satellite observations. Journal of Climate, 16(21), 3498-3510.

Guillevic, P., Göttsche, F., Nickeson, J., Hulley, G., Ghent, D., Yu, Y., Trigo, I., Hook, S., Sobrino, J.A., Remedios, J., Román, M. & Camacho, F. (2017). Land Surface Temperature Product Validation Best Practice Protocol. Version 1.0. In P. Guillevic, F. Göttsche, J. Nickeson & M. Román (Eds.), Best Practice for Satellite-Derived Land Product Validation (p. 60): Land Product Validation Subgroup (WGCV/CEOS), doi:10.5067/doc/ceoswgcv/lpv/lst.001. https://lpvs.gsfc.nasa.gov/PDF/CEOS_LST_PROTOCOL_Oct2017_v1.0.0.pdf

Overland, J. E., & Guest, P. S. (1991). The Arctic snow and air temperature budget over sea ice during winter. Journal of Geophysical Research: Oceans (1978–2012), 96(C3), 4651-4662.

Wiese, M., Griewank, P., & Notz, D. (2015). On the thermodynamics of melting sea ice versus melting freshwater ice. Annals of Glaciology, 56(69), 191.

Yu, Y., Rothrock, D. A., & Lindsay, R. W. (1995). Accuracy of sea ice temperature derived from the advanced very high resolution radiometer. Journal of Geophysical Research: Oceans (1978–2012), 100(C3), 4525-4532.

---

## Author Comment (AC1) · 16 Jan 2018

General comments:

The authors present 40 days of near-surface air and surface skin temperature measurements at Summit station Greenland. They analyse the datasets with focus on identifying the timing and magnitude of near-surface thermal inversion. They compare their best measurements of surface skin temperature, from IR upward radiation, to the latest MODIS Land Surface Temperature (LST) and find smaller bias than in other studies validating that product.

The skin temperature is a major variable in the surface energy balance of snow and ice surfaces and has drawn little attention so far. Product such as MODIS LST is a valuable tool for quantifying that variable but needs ground validation before it is used for the whole Greenland ice sheet. The authors therefor address an important question in a well written study using adequate tools and methods.

However, the study suffers at the moment from the lack of temporal and spatial coverage. Indeed one can ask if these 40 days of observations are representative of the surface conditions on the ice sheet and adequate to validate the MODIS LST as a whole. Especially, it is unfair to compare a 40-days comparison to previous studies that all use multiyear or spatially distributed datasets. As far as I know, NOAA can provide suitable data (IR skin temperature) for a longer period at Summit and PROMICE stations for sites at lower elevation.

> *We agree that these 40 days of observations are not representative spatially or temporally of the Greenland Ice Sheet as a whole. However, we decline to include NOAA data or PROMICE data in this paper for the following reasons: Both of these datasets do measure upwelling longwave radiation, and the PROMICE datasets calculate a skin temperature by assuming blackbody radiation properties of snow and ice surfaces. However, it's a very wide band that is used (4.5 to 42 µm), whereas our IR sensor uses the band from 8-14µm to calculate temperature. MOD11 uses bands around 11µm and 12 µm to calculate skin temperature, which is much more similar to the sensor we use to measure IR skin temperature. Furthermore, the PROMICE stations are all at ice sheet margins. While it would be very interesting to use this data to compare to MOD/MYD11, our study is focused on the near-surface phenomenon in the dry snow zone. Comparison of skin temperature, air temperature and MODIS surface temperature at PROMICE stations at the ice sheet margins would make for an excellent future study. While our dataset may not be very long, we have strong confidence in the quality of our data, as the station was maintained fairly closely and was not left alone for more than 1 month. PROMICE stations are not as frequently maintained. Also, though our data is not representative of all radiative conditions due to summertime measurement in constant sunlight, there is still a large spread in the temperatures measured (from -30 C to -5 C).*

Additionally the study could also benefit from more in-depth definition and discussion of all the measurements (f.e. how is calculated MODIS LST, effect of surface emissivity and of the air below the sensor for IR radiation measurements . . .) and concepts (what is skin temperature, how is defined thermal inversion, which indicators and metrics are used to describe it, which are its drivers. . .) that are being used.

> *We have added further discussion of the topics you list throughout the manuscript.*

Eventually the use of unshielded thermochrons and thermocouples is highly sensitive to radiation (even more during midnight sun period) and should not be used without appropriate segregation of the erroneous periods or correction of radiative heating. At the end of the study, the authors even discard these measurements and only compare MODIS to IR surface radiation. I would recommend greatly minimizing the description of that data and any conclusion that derives from it (f.e. statement regarding inversion in the 5 cm above the surface).

> *We have decided to remove the thermochron and thermocouple data from the manuscript because the sensors were unshielded, and therefore we reconsidered the validity of the conclusions drawn from that data.*

Technical corrections:

l. 19: Define NOAA. Define again in main text.

*We have added these definitions.*

I.84: Please only mention the conclusions from Good (2016) relative to snowy surfaces.

*We have limited the discussion of Good (2016) to only snow covered sites and seasons.*

I. 100: "at but" word missing?

*Sentence was editing and the word "at" was removed.*

I. 105: It is said" Berkelhammer et al. (2016) discuss the impacts of the surface- based temperature inversions (with 2 m air temperature as the base)". I can see in Berkelhammer et al. (2016) that they have measurements down to 20 cm. Please adjust or justify.

*The measurements down to 20 cm are the isotope measurements. However, upon further review, they do measure surface temperature. It is unclear whether it is used or not in some of the calculations, but we have edited our description to reflect this new understanding: "Berkelhammer et al. (2016) discuss the impacts of the surface-based temperature inversions on boundary-layer dynamics, showing that the stability of the atmosphere prevents mixing and ultimately limits accumulation at Summit. These recent studies have investigated near-surface processes at Summit because of the importance of surface energy balance and turbulent snow/atmosphere exchange in climate monitoring and ultimately prediction of larger scale circulation and future change in ice mass balance. However, while some surface temperature measurements at Summit have been made (Berkelhammer et al. 2016), controls on surface temperature gradients in the lowest 2 meters of the atmosphere, which are most relevant for the remote sensing community and also have important implications for changing ice sheet dynamics, have not been explicitly studied at Summit, Greenland."*

I. 114: "There are", there is

*We edited this sentence to say "There are many different…"*

I. 119: The equation points at the various variables that need to be calculated before the surface temperature can be determined. Please give a brief description of how it is being done for MODIS LST and how accurate is that calculation.

*We have added more information about the MODIS LST. In lines 175-197, details of the MODIS surface temperature product are given.*

I. 126: "its products are chosen as the remote sensing product" rephrase

*Rewrote this to read: "The MODIS instrument produces widely-used land surface temperature (LST), which we will use as the remote sensing product for comparison in this work."*

I. 130: "A number . . ." here validation studies are described. It seems to be the same topic as the next paragraph so consider merging them. Also you discuss validation of MODIS products before you actually describe what these products are and how LST is being calculated. Consider rearranging paragraphs in this section.

*The former Introduction, Background, and Methods sections have been rearranged with these suggestions (and those of the other reviewer) in mind.*

I. 134: You mention the recurrent confusion between near-surface, 2 m air temperature and surface skin temperature. Please define your vocabulary at the beginning of the manuscript and then be specific every time temperature is mentioned. Here what does "ground surface temperature" refer to? Check other cases throughout the manuscript.

*We have edited the sentence to clarify here that we were referring to both 2m and skin temperature (depending on what data the relevant study used).*

I. 140: You mention a cold bias in Østby et al. (2014) but it is not reported in Table 1.

*Thanks for pointing out the oversight. The bias is now reported in Table 1.*

I. 144: "A bias in the data can obscure or alter trends within a dataset." Very general statement, consider removing.

*Statement has been removed.*

I. 145: "Furthermore, it is possible that. . ." at this point, you have not exposed any theoretical (potentially from how MODIS LST is being calculated) or empirical reasons to think that one of the 2 m or skin temperature would match better than the other to the MODIS product. Either add a justification or reference for this hypothesis or move it to what is being interpreted from your validation.

*We have edited the paragraph about the MODIS product to discuss the fact that it is indeed measuring the*

*skin temperature of the snow, which justifies this statement. This sentence was added: "In the measurements of snow, the resulting temperature is representative of the top several microns of the surface at the snow/air interface because of the penetration depth of radiation at that wavelength, so it is indeed a skin temperature (Warren and Brandt, 2008)."*

l. 148: What do you mean by "standard"?
> *We have removed that language.*

l.154: "Both MOD/MYD11 and the preliminary version of the MOD29 special product were compared to our in situ data". If both have been compared and validated then both should be presented. The conclusion that a LST product developed for land performs better than a LST product developed for ice on the Greenland ice sheet is a very important conclusion. Quantifying and localizing the errors in MOD29 could be important for future studies. You can also validate only MOD/MYD11 but then don't mention that you validated MOD29 but not show the result.

> *Validation of MOD29 is indeed important; however, we have removed our mention of it, and discussion of this product is left to future studies.*

l.156: In this paragraph, the authors should provide a clear and concise description on how LST (and all necessary variables such as water vapour, air temperature, emissivity, and cloud cover) are being calculated. It is important so that the reader can be reminded of the assumptions and uncertainties linked to the calculation and to which level of accuracy can be expected from the product.

> *We have added some details to this section as you suggest, especially in the uncertainties linked to the calculation. We feel that this section already has much more detail than other similar studies (Westermann et al. 2012, Ostby et al. 2014, Shuman et al. 2014, etc.) and the interested reader can seek further detail of the MODIS calculations in the cited literature.*

l. 160: "feature" not clear what it refers to.
> *We have rewritten this sentence to improve clarity. "Calculation of surface emissivity is important in this product because MOD/MYD11 is a global product that estimates land surface temperature on all types of land cover types."*

l. 161: "Over snow and ice, this. . ." Not very clear, please rephrase.
> *We have rewritten this sentence to improve clarity: "Because this study focuses on consistently snow-covered land, there was not significant variability in the emissivity; in band 32 the emissivity is 0.990 for each data point, and in band 31, the emissivity fluctuates between either 0.992 or 0.994."*

l. 165: In this paragraph, please quantify the difference between C5 and C6. More especially, how much of the cold bias seen in validation studies can be explained by the defects of C5 products?

> *We conducted our comparison of in situ IR skin temperature to the C5 product to answer this question. We found a mean difference of 0.2°C between C5 and C6 in our study time and location, where the C6 temperatures are slightly higher than C5. This has been added to the manuscript and figures have been added to the supplement.*

l. 171: "decrease in measured brightness temperatures" define brightness temperature or use vocabulary previously defined. Does this sentence imply that the corrections in C6 would lead to even colder bias if compared to the previously mentioned validation studies?

> *We have edited this section and added further discussion of C6 vs.C5 and figures to demonstrate the differences in the supplement.*

l. 210: "the pixel that has the minimum distance" a pixel is an area so the station should be located within one at all time. Provide pixel size at some point in the manuscript.

> *The pixel is an area, but it is defined by central latitude/longitude point, so we use this minimum distance to find out which pixel our site is in. Pixel size is 1km x 1 km, and this information has been added to the manuscript.: Within each swath, we find the 1 km x 1 km square pixel that contains our measurement site by minimizing distance between pixel nadir point and our in-situ measurement site.*

l. 213: "the nonsynchronicity may introduce some error to the comparison" It seems that the IR radiation comes is recorded at 5 min interval and 2 m air temperature at 1 min interval so this nonsynchronicity error could be removed by taking the measurements that are within few minutes of the MODIS acquisition time. Even if it is random noise, removing that error could potentially show better match of MODIS LST with ground measurements.

> *The IR skin temperature data is taken every 5 minutes, but what is saved in the datalogger is only the 30*

*minute average (so that storage space could be minimized), so unfortunately we do not have 5 minute data for the IR skin temperature.*

l. 246 "Several different" redundant

*Edited to read: "Several types of sensors…"*

l. 247 "in order to compare this study to previous [. . .] studies" I assume the main goal was to validate MODIS, comparing to other validation studies comes after. Maybe rephrase.

*This is a good point. As we have removed the thermochron and thermocouple data due to data quality concerns, this sentence is no longer in the manuscript.*

l. 255 "In Koenig and Hall (2010) . . ." the following sentences should be in the methodology where the measurements and their expected limitations are being presented. Additionally Hall et al. (2015) seem to conclude that unshielded thermochrons are subject to measurement error. So why using/presenting that data at all? The use of unshielded thermocouple is subject to the same issue.

*We have moved this section to the methods. We have decided not to include the thermocouple and thermochron data.*

l. 270: "differences are much higher at lower wind speeds" Unfortunately, conditions favourable to inversions are also the one enhancing radiative heating of both thermochrons and thermocouple: low wind speeds at 2m imply even lower wind speeds at the surface and will hamper ventilation of the sensors leading to sustained radiation absorption even with low sunlight. Better information should be given to ascertain that this difference is due to inversion.

*We decided not to include the thermocouple and thermochron data because we were unable to account for the issues associated with the unshielded temperature measurements and because their inclusion detracted from the central focus of the paper.*

l. 274 "similar" quantify l.275 "larger" quantify

*To address the previous 2 comments, the following edits were made: "While 2 m air temperature and IR skin temperature are similar during peak solar irradiance (Figure 5), with the mean difference in temperature being -0.32°C when incoming solar radiation is greater than 600 W m$^{-2}$. There is a larger difference between the two during the night-time, with 2 m air temperature higher than skin temperature by an average of 2.4°C when incoming radiation is less than 200 W m$^{-2}$."*

l. 280 "increased discrepancy" seems redundant to what is said in the previous sentences. Also quantify here.

*We have added the following quantification: "As the inversions appear diurnal in nature, the measurements are quite similar at higher temperatures (above -10°C, mean difference is -0.16°C), but at lower temperatures, there is increased discrepancy between 2 m temperature and snow skin temperature (below -20°C, mean difference is 3.5°C)."*

l. 281 "most frequently" quantify.

*We have added the following quantification: "There is a clear skew in the histogram, indicating that 2 m air temperature is most frequently higher than skin temperature (in 68% of measurements). This is true in both clear and cloudy sky conditions, where the percentage of measurements for which air temperature exceeds skin temperature is 70% in clear sky and 65% in cloudy sky."*

l.289 The following paragraph brings in discussion about MODIS surface temperature when we are still in the ground measurement section. Consider moving to the next section.

*We see your point, but this is not a comparison of our data to MODIS temperatures, so we think it still fits best within this section.*

l. 294 "Previous studies acknowledge that near-surface stratification may be part of the cause of the discrepancy" this is very important and has not been mentioned clearly in the introduction. How did these studies arrive to these conclusions?

*More information on this has been added in the introduction: Shuman et al. 2014 acknowledge that differences between 2 m air temperature and skin temperature caused by inversions could cause bias in their comparison to MODIS, but at the time there was insufficient data to suggest whether inversions would persist in central Greenland and in the very near surface.*

l. 311 "In the summer. . ." and following. Can you be more quantitative, f.e. binning observation into night and day time and showing that the mean differences in each group are statistically different. Maybe also find a cut off value in radiation below which you have significantly stronger inversion.

*We have added more quantitative analysis as follows: "The mean differences reported above as -0.16±0.88°C*

*when temperatures are above -10°C and as 3.5±2.4°C when temperatures are below -20°C. A paired t-test shows that these means are not equal to one another with a p-value of less than 0.001."*

l. 319 You are just repeating what is said about Good (2016) in the introduction.

*We have removed the discussion of Good (2016) as it did not add anything new.*

l. 322 not clear what "median difference" with a +/- sign refers to.

*Upon further reading of Good (2016), I do not think the comparison to the "median difference" was relevant.*

l. 332 Can you specify what is the average solar radiation and zenith angle during that time.

*This is no longer relevant as we have removed the thermochron data from the study because we decided that the measurements were flawed since the thermochrons were not shielded.*

l. 335 Since you aim at validating MODIS LST, better or worse match with it is not a good reason for discarding ground measurements. Consider quantifying the error of the thermochrons using IR temperature instead. If these measurements were not considered reliable enough to validate MODIS LST, why are they being used to quantify the thermal inversion earlier in the study?

*We have removed the thermochron data from the study because we decided that the measurements were flawed since the thermochrons were not shielded.*

l. 335 please give mean bias along with RMSE

*We have removed the thermochron data from the study because we decided that the measurements were flawed since the thermochrons were not shielded.*

l. 350 please give mean bias along with RMSE

*The mean bias was already reported here. "This is also evident in Figure 10, where MOD/MYD11 products combine to yield and RMSE of 1.6°C (n=374) when compared with IR skin temperature, and there is a mean bias of 0.7±1.4°C."*

l. 356 please state at which viewing angle the error is maximal and quantify that maximum error

*We have indicated the range of viewing angles and the slope of the fit line. To indicate the maximum error and the angle at which it occurs may be misleading as there are some points that appear as relative outliers. Linear regression provides a better way to consider these errors.*

l. 360 Why not comparing MODIS to the thermocouple data that you also presented earlier? If, as for the thermochrons it is considered as erroneous measurements it should also be stated and in that case it shouldn't be used for *quantifying* near surface inversion earlier in the study. Also consider comparing MODIS with the 2 m temperature over the study period and see if you find same bias as previous validation studies using 2 m temperature.

*We have removed thermochron and thermocouple data from the manuscript. We have added a comparison of 2m temperature to MODIS temperature. The new paragraph is as follows: "While we do not believe that 2 m air temperature is a good proxy for skin temperature, for demonstration purposes, we have compared the 2 m air temperature measurements to the MOD/MYD11 product in Figure 10b. In doing so, we find an RMSE of 3.1°C and a mean bias of 1.9±2.5°C (n=374). This results in a similar RMSE to Shuman et al. (2014) of 3.5°C, though the mean bias of our comparison is slightly less than their bias was at 3°C. This comparison further illustrates the importance of using skin temperatures in MODIS validation studies. Shuman et al. (2014) were unable to conclusively say that any of their bias was a result of using 2 m air temperature instead of skin temperature, and in fact they did not think it was likely that any inversion effects would cause the gradually increasing bias with decreasing temperature because there was insufficient research on the presence of near-surface inversions in the dry snow zone in Greenland. The comparison of Figure 10a and 10b shows that at least in the summer, inversions were likely to have played a large role in their 2014 results."*

l.362 Along with the present discussion, it would be important to analyse the performance of MOD35 using the MMCR (false positive, false negative) to draw more broad conclusions (also using findings from Østby et al. (2014)) about when and where MODIS LST might be wrong because of cloud cover.

*We have added an analysis of false positives, true positives, false negatives, and true negatives into our discussion of cloud masking: "In comparing the MMCR data to the MOD/MYD11, we find that of the 1059 times that the site was within the field of view of the satellites in June and July of 2015, there were 585 instances when both MMCR and MODIS detected cloud cover, 288 instances when both MMCR and MODIS indicated clear sky. This indicates 82% agreement. There were 86 false negatives (where MMCR indicates clouds and MODIS does not) and 100 false positives (where MMCR indicates clear sky, and MODIS indicates clouds). Østby et al. (2014) also use in-situ cloud data to filter out MODIS surface temperatures that are impacted by the presence of clouds in their study in Svalbard. They found an overall false negative*

*rate of 17%, whereas our false negative rate was 8%. Their work shows that the MOD35 cloud mask performs more poorly in the winter than in the summer, so the difference in false negatives is likely due to more favourable conditions for effective cloud masking due to constant sunlight during our measurement period."*

l. 381 "5 cm nearest to the..." this conclusion comes from potentially heated thermo- couple. Needs to be mentioned.

*Thermocouple data have been removed.*

l. 387 "the lower RMSE is likely . . ." here you make a hypothesis when you could actually show it. Is your RMSE greater when comparing MODIS to 2 m temperature? Please discuss mean bias along with RMSE.

*We have added the comparison of MODIS to 2 m temperature and included the data in the paper. In the conclusion, we have edited this sentence as follows: "The lower RMSE and mean bias is likely a result of measuring the skin temperature using an IR instrument directly (instead of using 2 m air temperature, which resulted in an RMSE of 3.1°C and a mean bias of 1.9°C)."*

The authors present results from a new field campaign near Summit Station, Greenland conducted between 8th June and 18th July 2015. This campaign data is used to investigate near-surface temperature inversions at the site and validate MODIS MOD/MYD11 Collection 6 products. Overall this is an interesting paper which presents original results from a short field campaign as well as from validation of MODIS collection 6 products over Greenland. The manuscript is generally well-written and the methods used are appropriate.

However, I feel that there are issues which need to be addressed by the authors in redrafting this manuscript. There are also sections of this manuscript which would benefit from a tightening of prose and improvement of structure. Both are elaborated on in the specific comments.

Specific Comments:

It was hard to ascertain the new and original contributions that this manuscript provides to current scientific understanding at first. Outlining these in the introduction or similar would aid reader understanding.

> *We have edited the final paragraph of the introduction to make our contributions more clear. The final paragraph of the introduction now reads as follows:* "*We use our original dataset to determine how summertime meteorological conditions impact near-surface inversions (beneath 2 m height) on the ice sheet at Summit. Furthermore, we provide a validation of MODIS land surface temperatures, and show that the use of 2 m air temperature for MODIS validation is not recommended due to the presence of near-surface inversions. Lastly, we use in situ cloud data to show that the accuracy of the MODIS surface temperature product could be improved through stricter cloud masking.*"

Line 14: suggest "can be assessed" or similar rather than "are assessed" as satellite derived temperatures over land are not in common use for this yet.

> *We have edited this sentence as you suggest.*

Line 53-54: The section starting "however, satellite remote sensing" should mention that the focus of this manuscript is on thermal infrared remote sensing, rather than microwave remote sensing, of surface temperatures. It should also be mentioned that thermal infrared remote sensing observations are affected by cloud cover. The sentence currently gives the impression that the satellite derived surface temperatures are spatially and temporally complete.

> *We have added this information as you suggest. This section now reads as follows:* "*In addition, thermal infrared satellite remote sensing provides the opportunity to collect surface temperature with large spatial coverage and sub-daily to weekly temporal resolution, depending on cloud conditions. In this study, we will focus on the MODerate-resolution Imaging Spectroradiometer (MODIS) thermal infrared land surface temperature (LST) product.*"

Line 60-62: "inversions. . . which may cause a disparity between the 'surface' temperature at 2m and the actual skin temperature of the snow surface". This sentence currently gives the impression that there is uncertainty about whether inversions cause a disparity between skin and 2 m air temperatures over snow and ice. Yet, the references cited in Section 2.1 note both the presence of inversions in these areas and their effect on temperature stratification. In a related issue, the manuscript could be read as suggesting that the issue of inversions (or other causes of disparity between 2m and skin temperatures) in validation of surface temperatures from thermal infrared remote sensing has not previously been considered. However, various previous studies have noted the issue of the difference between skin and 2 m air temperature over snow and ice surfaces in relation to remote sensing of surface temperatures and their validation. The effect of inversions on the difference between skin and 2 m air temperature over snow and ice surfaces has been noted by e.g. Comiso et al, 2003; Yu et al, 1995. There are also other reasons why differences may be seen between these two temperatures over cryospheric surfaces under clear sky conditions e.g. ice crystal precipitation (Yu et al, 1995) and latent heat effects (Wiese et al, 2015). As a result it is recommended that validation of satellite surface temperatures over land areas is done with situ surface temperatures (ideally from ground-based radiance

measurements) if possible (Guillevic et al, 2017). In light of these previous studies and recommendations, this sentence at least should be rewritten and references included in the manuscript to give credit to previous work looking at the issue of the difference between skin and 2 m air temperature over snow and ice surfaces in relation to validation of remotely sensed surface temperatures.

*Thank you for sharing these references. We absolutely want to give credit to these authors and have included this information in the introduction, and in section 3.2.1.*

Lines 66-73: Does question b) refer to the specific in situ sensors you use, or to the use of in situ skin temperature data for validation in preference to air temperatures? I could not find these questions referred back to in the results or conclusions. Would suggest removing these or referring back to them later in the manuscript.

*I have replaced these questions with statements about the main contributions of our work (relevant to comment above).*

Section 2: The content of the background section was informative and interesting. However, in combination with the introduction and methods I found the structure of the manuscript a little hard to follow here. This section would benefit from a rewrite, or a restructuring. Some suggestions follow. The content of the surface temperature inversion section would provide a nice introduction to the issues raised in lines 55-65. The Remote Sensing of Surface Temperatures section mostly focuses on MODIS rather than remote sensing of surface temperatures more generally. The content relating to MODIS products could be moved to Section 3.2 or the section could be renamed.

*We have restructured the Introduction/Background/Methods sections to improve the flow and clarity. The background on inversions was moved into the introduction, and the background on remote sensing was moved into the introduction and methods sections.*

Line 215 to 220: these metrics are in very common use so the equations do not need stating unless there are notable differences from how they are commonly applied

*We have removed the equations from the manuscript.*

Line 241: There is very little discussion relating to figure 3. Reconsider including this figure or provide more discussion of the results shown.

*We have added more discussion of this figure in the first paragraph of section 3.1. We believe it provides a nice context for the comparisons that follow.*

Section 4.1: I found this section quite hard to read and understand. Please rewrite or restructure. Also, please include some values for the differences (bias, RMSE, etc.) noted between the in situ sensor surface temperatures to provide context to previous study comparisons (lines 304-329).

*With the removal of the thermochron and thermocouple data, we have restructured this section and included bias and RMSE between 2m air temperature and IR skin temperature.*

Lines 246-260: Given the issues noted with the thermochrons, please reconsider including these or provide some comment on whether these sensors are suitable for measuring surface temperature. If the authors decide to retain the thermochron analysis, in Figure 4 the difference between the thermochrons and the other sensors in- creases noticeably after day of year 167 but before they are buried by snow. Please provide some comment on this.

*We have decided to remove the analysis with the thermochrons from the paper because the sensors were unshielded and so the quality of the data was uncertain.*

Lines 266-267: "The measurements show". Ambiguous as to whether this refers to the measurements affected most by solar heating or other observations.

*No long relevant as we have decided not to include these measurements in our manuscript.*

Lines 304-318: Did Hall et al. 2008 use the same infrared radiometer as in the study detailed in the manuscript? Table 1 suggests this study only looked at 2 m air temperature.

*Hall et al (2008) states that 'surface temperatures were derived from two positions on an AWS mast' and also that Everest 4000 4ZL TIR sensor that was placed 50 cm above the surface to record the surface temperature assuming a snow emissivity of 0.99. They did not use the IR data for MODIS validation.*

Line 317-318: Sentence is ambiguous as to whether the need for future studies refers to the results of your study specifically, or in general.

*Sentence has been edited as follows: "Future studies beyond our analysis that incorporate all seasons are needed to investigate this discrepancy and determine conditions under which 2 m air temperature is, or is not,*

*a good proxy for snow skin temperature."*

Lines 323-324: Is there publicly available snow depth information at these sites to address this question if it is not included in the paper?

*Using satellite imagery, I was able to determine that these sites are not continuously snow covered in the summer. As a result, we are not including comparisons to their summer results.*

Lines 324-329: Why was the thermocouple data not compared to MODIS? If there is a reason this should be stated.

*The thermocouple data was not compared to MODIS because it is not a skin temperature, as MODIS is. Furthermore, issues of thermocouple heating have caused us to remove this dataset from the manuscript.*

Lines 353-359: Do the authors have any thoughts on what else could be causing the remaining differences? If so please include this.

*We believe that a stricter cloud mask would improve the difference, which we address in the next section. Furthermore, non-synchronicity between ground based measurements and MODIS measurements may be to blame for random noise. We have included the following sentence: "We believe that the differences may be due to insufficient cloud masking and perhaps to imperfect synchronicity of measurements, where in situ skin measurements represent an average of 30 minutes but the MODIS measurement represents a shorter time window."*

Lines 366-368: I think the authors say "improving the cloud mask" when they mean increasing cloud masking strictness which may improve the product but also overflag cloud so that there is loss of un-cloud contaminated data ("reduce the amount of measurements available"). Data loss due to cloud masking, assuming that the pixels re- moved are genuinely cloud contaminated and the cloud masking is therefore accurate, is not a problem as these pixels will not contain sensible infrared surface temperature estimates. The issue is when there is significant over-flagging of cloudy pixels, leading to loss of non-cloud contaminated data, due to an increase in cloud masking strictness. If so these sentences (and lines 395-396) should be re-written.

*Yes, you are correct, and we have edited our text to improve clarity. "In determining the strictness of the cloud mask used, there is a trade-off due to the need to mask out all cloud contaminated pixels but not overflag data, which results in the generation of false positives and removes pixels that were in fact clear."*

Line 384: Do not use the word "correct" here as this suggests that MODIS is perfectly accurate, when actually MODIS data (and indeed any observation) will not measure the true (generally unknown) value of the surface temperature. There are always biases and uncertainties when measuring.

*This is a good point. The wording has been edited as follows: "indicate that the MODIS data has only a very slight cold bias (-0.7°C)"*

Lines 390-391: Sentence unclear in meaning. Are the authors suggesting using MODIS data and in situ 2 m air temperature to study inversions? If so, are the biases between the MODIS and in situ skin temperatures understood adequately to allow such a study? Lines 353-359 suggest not.

*This is indeed what we were suggesting. We think that with further work, these biases would be adequately understood to allow such a study. The text has been edited as follows: "Furthermore, the validation presented in this study of the strong correlation between MODIS surface temperature and snow skin temperature in the summer lays a groundwork for inversions to be studied more extensively in locations where 2 m air temperature is currently measured."*

Table 1: This table could do with a little restructuring and/or reduction of text as it is currently a little difficult to read and understand. Also, if this is for studies over land ice only please include this information in the caption.

*The table has been edited to reduce text and hopefully improve clarity. The caption has been edited to clarify that this data is for "snow-covered regions."*

Technical Corrections:

Line 47: "for understanding ice sheet..." rather than "of understanding ice sheet..."?

*Correction made.*

Line 53: remove extra space in "Fausto et al., 2012) ; however"

*Correction made.*

Line 534: remove curly brackets

*Correction made.*

Figure 1: The location dot is a little small. The north arrow is also a little difficult to see.

*Correction made.*

Table 1: missing "n=" on last row.

*Correction made.*

References:

Comiso, J. C. (2003). Warming trends in the Arctic from clear sky satellite observations. Journal of Climate, 16(21), 3498-3510.

Guillevic, P., Göttsche, F., Nickeson, J., Hulley, G., Ghent, D., Yu, Y., Trigo, I., Hook, S., Sobrino, J.A., Remedios, J., Román, M. & Camacho, F. (2017). Land Surface Temperature Product Validation Best Practice Protocol. Ver- sion 1.0. In P. Guillevic, F. Göttsche, J. Nickeson & M. Román (Eds.), Best Practice for Satellite-Derived Land Product Validation (p. 60): Land Product Validation Subgroup (WGCV/CEOS), doi:10.5067/doc/ceoswgcv/lpv/lst.001. https://lpvs.gsfc.nasa.gov/PDF/CEOS_LST_PROTOCOL_Oct2017_v1.0.0.pdf

Overland, J. E., & Guest, P. S. (1991). The Arctic snow and air temperature budget over sea ice during winter. Journal of Geophysical Research: Oceans (1978–2012), 96(C3), 4651-4662.

Wiese, M., Griewank, P., & Notz, D. (2015). On the thermodynamics of melting sea ice versus melting freshwater ice. Annals of Glaciology, 56(69), 191.

Yu, Y., Rothrock, D. A., & Lindsay, R. W. (1995). Accuracy of sea ice temperature derived from the advanced very high resolution radiometer. Journal of Geophysical Research: Oceans (1978–2012), 100(C3), 4525-4532.

*All of these references have been added except for Wiese et al., which was a very interesting paper, but did not seem directly relevant to our study.*

---

## Author Response (AR2)

**Reviewer #1**

General comments:

The revised manuscript answers satisfactorily to my previous comments and is now clear and nicely structured. I have just few more comments regarding the writing or elements that are still missing the analysis.

Specific comments:

It should be mentioned somewhere that this study does not include surface temperatures close to melting point. It means that the timing and duration of melt detected by MODIS LST is still not validated with surface measurements. Potentially, it was done with 2m air temperatures as proxy in previous studies, but your study shows that this proxy is not performing well.

> *To address this point, we have added the following text in section 3.2.1: "Across the range of temperatures in the study (approximately -30°C to -5°C), the agreement is consistent. Due to the conditions that occurred over our study period, we did not capture temperatures near the melting point, as surface melt is very rare at Summit, or at the lower temperatures common to winter conditions at Summit."*

> *And we have added the following to the conclusions: "In the future, we plan to extend studies of this type to longer spans of time to determine if these results also are representative of lower temperatures and to capture higher temperatures as well, providing further validation of the MODIS surface temperatures near the melting point."*

l. 24 "during summer months": You do not have enough data to conclude on 'summer' conditions in general or even for summer 2015. Please consider more conservative 'during our study period' or similar.

> *We have edited the wording as you suggest.*

l.24 "after additional cloud masking": Please also give the performance with standard MODIS cloud masking as it is the one future users of MODIS LST will use.

> *We have added the RMSE and mean bias before additional cloud masking to the abstract.*

l.42 "Furthermore, the energy balance …." The following sentences are unclear and fail to link the general context given previously to your studied topic: surface temperature. L.48 In the same way you state that surface temperature is a critical variable but do not go into details. Consider being more specific about how and why surface temperature is important (tracking melt, heat exchange with atmosphere, firn warming/cooling …) to show better the motivations of the study.

> *To address these points, we have edited the first paragraph of the introduction, shifting the focus back to surface temperature (and hoping to clarify the link between surface temperature and surface energy balance) and providing more direct motivations for the importance of surface temperature.*

l.65 "is defined by…" very bad definition of the process at the core of your study: Inversion is not defined by measurements. It can be detected with them (and not only at two levels). The sign of the temperature difference also matters. Please rephrase.

> *We have edited this paragraph to reflect your suggestions. We meant that inversions could be detected through measurements at two (or more) heights, but you are right that language was not clear. The new structure and edits improve upon the previous version.*

l.74 "… these two temperatures generally agree well …" seems like Good (2016) did not find any inversion over snow? Is that right? Then more explanation about why some studies detect inversion and some do not would be useful.

> *The reduced amplitude of the diurnal 2 m air temperature implies an inversion at night. I have added this to the description: "Good (2016) presents measurements of skin temperature and 2 m air temperature, and finds that at polar sites, during snow-covered seasons in fall, winter, and spring, these two temperatures generally agree well, with the caveat that there is a reduced amplitude of diurnal cycle temperatures at 2 m, which would imply a temperature inversion during the night and a temperature lapse during the day."*

l.76 This paragraph is redundant as it defines again thermal inversions and its drivers (l.61 "In the polar regions…"), but in a much better way than previously. Consider merging with previous paragraph.

> *We have now rearranged the introduction and merged this paragraph into the previous paragraph.*

l.100 In the previous paragraph you already listed studies that detected and measured near-surface thermal inversion on the Greenland ice sheet. This paragraph shows that the same was done for Arctic sea ice which is less relevant for your application at Summit. Could be removed or relocated. As a general rule, it is clearer to go from the general (definition of inversion; measurements on land, seasonal snow and sea ice) to the specific (previous study of thermal inversion at Summit). Consider rearranging accordingly.

> *We have rearranged the introduction and moved the Comiso (2003) description to earlier in the introduction.*

l. 321 "As these variables …" Out of all the assumptions that were made (atmospheric conditions and compositions, bandwidth used by MODIS vs. bandwidth used by IR sensor ...) why would cloud masking and non-synchronicity be the main source of divergence? You stated that non-synchronicity would act as random noise, why would it now explain a small negative bias? From your result it is indeed visible that imperfect cloud masking is responsible for part of the negative bias. But why is that? Can you please explain and interpret?

> *We have added more information to this section. We don't think that the non-synchronicity would cause a negative bias, we merely want to acknowledge it as a source of error, as there are both positive and negative residuals between the IR surface temperature and the MODIS surface temperature.*

> *"As these variables do not explain much of the difference, other potential sources of the discrepancy may be insufficient cloud masking (discussed in the following section), assumptions within the MODIS algorithm to determine atmospheric composition and properties, or imperfect synchronicity of measurements, where in situ skin measurements represent an average of 30 minutes but the MODIS measurement represents a shorter time window. Previous studies have shown that cloud masking limits the accuracy of surface temperature products in snow-covered areas (Westermann et al. 2012; Hall et al. 2004). In particular, the presence of clouds can lead to a negative bias because clouds can be misinterpreted as snow surface, and they often have lower temperature than snow surface temperatures. Yu et al. (1995) suggest that ice crystal precipitation present during inversions may also cause differences between in situ and satellite skin temperatures, though they caused a warm bias rather than a cold bias."*

l.338 "This indicates …." More discussion/interpretation is needed from this comparison. Are these numbers satisfactory? Would you recommend changing the MOD35 threshold? Or would that lead to too much false negative. How to choose?

> *We would recommend considering improvements to the MOD35 algorithm. We have added the following text to our manuscript:*

> *"Our results indicate that improvements to the MOD35 cloud mask would be beneficial. A stricter threshold would ensure that fewer cloud-covered pixels are included in the surface temperature dataset but would also likely lead to more false positives. Making this threshold decision may depend on the level of error that is acceptable given the analysis at hand. The ideal improvement would not be merely to change the threshold value, but to continue to improve cloud detection algorithms, which is continually done with each MODIS collection iteration (e.g. Riggs et al., 2017)."*

**Reviewer #2**

Introduction: It could benefit from some tightening of prose as it currently fairly long for an introduction and there is some repetition of explanation, but the content is good and it reads well.

> *Along with some suggestions from the other reviewer, we have made edits to the introduction to reduce repetition and tighten the discussion.*

Line 152: It was a little unclear on first reading that the IR measurements are taken every 30 minutes. For clarity it might be worth mentioning this temporal sampling interval in the previous paragraph (where you describe the IR measurements).

> *Thanks for pointing this out. The sampling interval had been in a previous draft of the manuscript, but accidentally got removed. We have added that information back in. "
[revised manuscript text omitted]